# Fatty Acids of Echinoderms: Diversity, Current Applications and Future Opportunities

**DOI:** 10.3390/md21010021

**Published:** 2022-12-27

**Authors:** Natalia V. Zhukova

**Affiliations:** National Scientific Center of Marine Biology, Far Eastern Branch, Russian Academy of Sciences, 690041 Vladivostok, Russia; nzhukova35@list.ru; Tel.: +7-423-231-0937; Fax: +7-423-231-0900

**Keywords:** fatty acids, polyunsaturated fatty acids, nisinic acid, biological activity, Echinodermata, Crinoidea, Ophiuroidea, Asteroidea, Holothuroidea, Echinoidea

## Abstract

The phylum Echinodermata comprising the classes Asteroidea, Ophiuroidea, Echinoidea, Holothuroidea, and Crinodeia, is one of the important invertebrate groups. Members of this phylum live exclusively in marine habitats and are distributed in almost all depths and latitudes. Some of them, such as sea urchins and sea cucumbers, are commercially valuable and constitute a major fishery resource. Echinoderms are increasingly recognized as a unique source of various metabolites with a wide range of biological activities. The importance of dietary polyunsaturated fatty acids, such as eicosapentaenoic acid, in human health has drawn attention to echinoderms as a promising source of essential fatty acids (FAs). Extensive information on the FAs of the phylum has been accumulated to date. The biosynthetic capabilities and feeding habits of echinoderms explain the findings of the unusual FAs in them. Certain common and unusual FAs may serve as chemotaxonomic markers of the classes. The main goal of the review was to gather the relevant information on the distribution of FAs among the echinoderm classes, describe the structures, distribution, biosynthetic pathways, and bioactivity, with an emphasis on the FAs specific for echinoderms. A large part of the review is devoted to the FAs derived from echinoderms that exhibit various biological activities promising for potential therapeutic applications.

## 1. Introduction

The Echinodermata is a phylum of marine invertebrate animals found exclusively in marine habitats [1]. The name Echinodermata is derived from the Greek words *ekhinos* (meaning spiny) and *derma* (skin). Approximately 7000 species of extant echinoderms have been described from all the oceans. At present, five classes are recognized in this phylum: Crinoidea (sea lilies and feather stars), Holothuroidea (sea cucumbers or holothurians), Echinoidea (sea urchins and sand dollars), Asteroidea (sea stars or starfish), and Ophiuroidea (brittle stars) (Figure 1). Echinoderms are characterized by radial symmetry; many of them have several arms, mostly five or more, radiating from the central part of the body. These animals are distinguished also by having tentacle-like structures referred to as tube feet with suction pads. Their body and arms are covered by the spiny skin.

Members of this phylum are distributed in almost all latitudes, environments, and depths, from the intertidal to the deepest, abyssal zones of the ocean. They can be found on the seabed in every climate zone, from the tropics to the polar regions. Echinoderms constitute an important component of the invertebrate fauna in the ocean [1]. They numerically dominate many marine communities and play a variety of ecological roles, being important links in food webs and key components of grazing communities. Echinoderm species occupy various subtidal substrates, from stony to fine silt sediments, are most commonly found on sandy bottoms in coastal waters, and also dominate the megafauna of the abyssal sea bed [2].

Some echinoderm species, in particular holothurians and sea urchins, are intensively harvested by commercial fisheries in several countries. Sea urchins are valued for their gonads, which are a highly prized seafood product and a delicacy in many regions of the world, especially in Asia and Europe. These animals are an important human food item in the Caribbean. In the Mediterranean countries, there is a long tradition of harvesting and eating sea urchins. In particular, Italy has Europe’s largest market for *Paracentrotus lividus*, with the annual consumption of this sea urchin species in the country reaching about 2000 tons. Holothurians are abundant in waters of the Asia-Pacific region, where many species are harvested for human consumption and some are raised in aquaculture. Harvested Japanese sea cucumber, *Stichopus (=Apostichopus) japonicus*, is variously referred to as trepang, namako, bêche-de-mer, or balate. Being considered a perfect seafood of high nutritional value, it is widely consumed in China, Japan, South Korea, and other southeast Asia countries. 

Echinoderms have been poorly studied with regard to their medicinal values, except for sea cucumbers, that are widely used in Chinese medicine as an effective remedy against several diseases [3]. In traditional medicine, e.g., in Brazil, echinoderms such as sea stars, sand dollars, and sea urchins are used in therapies against asthma, alcoholism, bronchitis, diabetes, and heart diseases. Tea from the powdered toasted starfish *Echinaster brasiliensis* is also a popular medicinal drink in this country [4]. The therapeutic potential of echinoderms is explained by their rich chemical diversity, which makes them a promising source of unique therapeutic agents for ethnopharmacology and natural products research [4,5,6].

Sea urchin larvae are widely used, mainly as model organisms, in developmental biology and ecotoxicology for their rapid growth and sufficiently large size. Many echinoderms possess outstanding regenerative capacities. The arm regeneration potential of echinoderms is being actively studied for understanding and treating neurodegenerative diseases in humans [7].

The Echinodermata is a rich source of bioactive compounds from different classes of natural substances [4]. Echinoderms have attracted the attention of scientists over the past few decades after a variety of their unique structures with promising biological properties were described. The phylum Echinodermata provides a wide range of opportunities for chemotaxonomic studies. It is a specific phylum of marine invertebrates that is reported to synthesize unique secondary metabolites playing an important role in the chemical defense of marine sessile and slow-moving organisms. Studies based on the screening of secondary metabolites produced by echinoderms are important for understanding their biotechnological potential. Representatives of the Echinodermata are of great interest for the research and development of bioactive natural metabolites. Since marine biodiversity reflects also the chemical diversity, the vast variety of species of this phylum offers a wide range of opportunities for discovering novel bioactive agents, through screening against various disease targets. Numerous reviews have been published on bioactive natural compounds derived from echinoderms. However, no reviews on echinoderm lipids and fatty acids (FAs) are currently available. 

The importance of polyunsaturated fatty acids (PUFAs) for human health and nutrition has prompted chemists, biochemists, and biotechnologists to focus their attention on FAs from marine organisms. Marine-derived FAs are extensively reviewed to elucidate their occurrence, roles, and for analytical methods [8,9]. The lipid chemistry of the Echinodermata has also been a subject of numerous attempts to discover new metabolites with intriguing biological activities. To date, there is an extensive body of information on the FAs of numerous members of the phylum Echinodermata, in particular edible species, are well as those studied and used in the human diet. A serious impetus in the study of FAs of marine organisms has been given by a significant progress in the methods of analysis of these components and an increasing interest in the compounds referred to as marine lipids, mainly polyunsaturated fatty acids (PUFA), having positive effects on human health due to their constituents, the highly valued n-3 PUFAs. 

The present review is based on a consideration of the most interesting advances in the study of echinoderm FAs. Thus, without covering the subject exhaustively, this review makes an emphasis on the most important factors and effects that determine the biodiversity of the echinoderm FAs, with focus on the crucial relationship of FAs with the biosynthetic capacities of animals and with their food spectrum. The major goals of this review were to illustrate the molecular biodiversity of FAs in echinoderms and the distribution of FAs over the classes of this phylum, as well as to identify the most important FAs with bioactive potential.

## 2. Echinoderms as a Source of Various Bioactive Substances

Marine invertebrates are a rich source of natural chemical compounds with various therapeutic potentials. As many as approximately a third of natural products of marine origin have been derived from echinoderms. Anthraquinones, pyrrole oligoglycosides, steroids, saponins, polyketides, fatty acids, sulphated polysaccharides, lipids, peptides, and terpenes are the major bioactive compounds isolated from echinoderms. To date, numerous comprehensive reviews have summarized the available data and provided the most complete information on the chemical structures, patterns of distribution, and biological activities of the vast variety of natural compounds derived from marine echinoderms over the past three decades [6,10,11]. This chapter briefly elucidates the importance of the echinoderm classes as an inexhaustible source for discovering novel drugs.

Crinoids, being the most primitive group of extant echinoderms, have unique chemical signatures. Interest in the chemistry of crinoid-derived natural product is increasing. Antibacterial, antioxidant, anti-diabetic, and anti-algal activities have been reported for two feather star species, *Comaster schlegelii* and *Himerometra robustipinna* [12]. Being sessile animals, crinoids use chemical defense mechanisms including toxic chemical compounds to protect themselves from predators. The toxicity can be attributed to a series of polyketide-derived, brightly colored, heavily oxidized quinones [10]. Crinoids produce secondary metabolites such as anthraquinones and naphthopyrones that possess various biological activities, including cytotoxicity against tumor cell lines [13], inhibition of NF-κB activation that is responsible for cancer development and inflammation [14], as well as anti-inflammatory and analgesic activities [15]. Naphthopyrones isolated from the feather star *Capillaster multiradiatus* inhibit the multidrug transporter ABCG2 that is involved in a key role in drug resistance [16], and exhibit moderate inhibition on in vitro HIV-1 replication in a T cell line [17]. Furthermore, cytotoxicity was reported for sulfated naphthopyrone derivatives against the LNCaP (prostate cancer) cell line and the SK-Mel-2 (melanoma) cell line [18]. Extracts from this feather star show antibacterial activity against several pathogenic bacteria [19]. In addition to secondary metabolites from crinoids, other structural classes have been identified, including the cerebroside CJP1 isolated from the feather star *Comanthus japonica*, which has antihepatotoxic properties [10].

Although Ophiuroidea is the largest class of echinoderms, studies on metabolites derived from brittle stars are relatively limited compared to those from other echinoderms. Nuzzo et al. consider several classes of secondary metabolites, such as carotenoids, gangliosides, brominated indoles, phenylpropanoids, several groups of terpenes, and steroids that have been isolated from brittle stars [20]. A methanol extract from *Ophiura albida* showed promising antioxidant and antimicrobial activities, probably due to their accumulated phenolic compounds. The extract also has an antidiabetic activity since it could significantly inhibit α-amylase and α-glucosidase involved in the digestion of starch and glucose production [5]. Two new sesquiterpenoids isolated from the brittle star *Ophiocoma dentata* have exhibited cytotoxic and anti-proliferative activities against the MCF-7 cell line and in Ehrlich tumor-bearing mice. The extract from this brittle star have also shown antimicrobial activities against several human bacterial pathogens [21]. The presence of sulfated steroids in brittle stars and starfish is an indicator of the phylogenetically close relationship between these two classes of echinoderms [22].

Holothuroidea and Asteroidea are the two major classes of echinoderms that have been explored to the greatest extent to find novel natural compounds [4]. Starfish are considered an extremely rich source of biologically active compounds, e.g., steroidal glycosides, steroids, anthraquinones, alkaloids, glycolipids and phospholipids. In a study conducted by Mohamed et al., saponin isolated from *Holothuria arenicola* exhibited significant antineoplastic activity [23].

Glycosphingolipids from starfish and feather stars have unique biological properties used in the treatment of dementia, osteoporosis, and diabetes and may become the seeds of preventive or therapeutic drugs for these illnesses [24]. Starfish and sea cucumbers cerebrosides show various types of biological activities that are important for their practical application in the human diet and in the formula of food supplements [25].

Echinochrome (EchA), a natural quinone pigment isolated from sea urchins, exhibits significant pharmacological activity and, therefore, it has been approved for use in human medicines, usually for the treatment of cardiopathies and glaucoma [23]. It is suggested as a drug to alleviate cytokine storm syndrome [26]. EchA can mediate cellular responses, acts as a radical scavenger, and activates the glutathione pathway. It decreases reactive oxygen species (ROS) imbalance, prevents and limits lipid peroxidation, enhances mitochondrial functions, and, most importantly, contributes to immune system modulation. EchA can regulate the generation of regulatory T cells, and inhibits pro-inflammatory IL-1β and IL-6 cytokine production, while slightly reducing IL-8, TNF-α, INF-α, and NKT, thus, correcting immune imbalance [26]. Organic extracts, as well as pure compounds, from sea urchins have anti-fungal, anti-parasitic, anti-inflammatory, hepatoprotective, anti-viral, anti-diabetic, anti-lipidemic, gastro-protective, and anti-cardiotoxic effects [27].

Echinoderms use the chemical defense mechanisms to protect themselves against microbial infections and, therefore, they represent a promising resource for finding effective antibacterial compounds. The results of a study of antimicrobial activity have shown marine echinoderms as a potential source of new types of antibiotics. They demonstrated stronger antibacterial effects than those from Porifera, Mollusca, Bryozoa, or Annelida [5]. Jebasingh et al. reported that ethyl acetate extracts from *Comaster schlegelii* and *C. multiradiatus* showed maximum antibacterial activity against such human pathogens as *Staphylococcus aureus*, *Bacillus subtilis*, and *Escherichia coli* [19]. Haug et al. mentioned a variety of antimicrobial compounds, including steroidal glycosides, polyhydroxylated sterols, naphthoquinone pigments, lysozymes, antimicrobial peptides, and complement-like substances isolated from echinoderms [28]. Lysozyme-like and hemolytic factors were also detected in the extracts [28]. Antimicrobial activity was recorded for extracts from *Astropecten irregularis*, *Luidia sarsi*, and *Ophiura albida* against the following human pathogens: *Staphylococcus aureus*, *Escherichia coli*, *Pseudomonas aeruginosa*, *Salmonella enterica*, and *Bacillus subtilis* [5]. A total of 22 species of echinoderms (one crinoid, three holothuroids, three ophiuroids, and 15 asteroids) have been reported to exhibit antimicrobial activity against bacterial pathogens [29]. Antimicrobial effects have been reported for extracts from various Echinodermata species such as the sea cucumbers *Holothuria leucospilota* [30] and *Cucumaria frondosa* [28], the sea urchin *Strongylocentrotus droebachiensis* [28], the sea star *Asterias rubens* [28], and *Ophiopholis mirabilis* [31].

The potential antidiabetic effect is evaluated via inhibition of α-amylase and α-glucosidase that convert dietary polysaccharides into monosaccaharides. All consequences associated with type 2 diabetes are mainly due to the elevated level of blood glucose in the body. Many studies have reported that echinoderms show significant antidiabetic activity. The ethanolic extracts from the feather stars *Comaster schlegelii* and *Himerometra robustipinna* have a moderate inhibitory effect on the enzyme α-amylase [12]. Extracts from three echinoderms *Astropecten irregularis*, *Luidia sarsi*, and *Ophiura albida* could significantly inhibit amylase and glucosidase involved in the digestion of starch and the glucose production [5]. Extracts from the sea urchin *Echinometra mathaei* showed significant anti-diabetic activity via inhibiting the α-amylase enzyme in a dose-dependent manner [32].

Several studies have shown that echinoderms are a promising source of potent antioxidants that protect organism from free radicals and ROS. They prevent lipid peroxidation and progress of many pathological states. Numerous studies have reported marked antioxidant activity of echinoderms. The ethanol extracts from the feather stars *Comaster schlegelii* and *Himerometra robustipinna* showed free radical scavenging activity, with the highest level of activity recorded from the latter species (*H. robustipinna*) [12]. Raj et al. refer to the species *Echinaster sepositus*, *Sphaerechinus granularis*, and *Arbacia lixula*, as well as the sea cucumber *Holothuria atra*, which demonstrates free radical scavenging properties and antioxidant activity in animal models in vivo, as a potential source of antioxidant compounds [12]. Viscera of the sea cucumber *Cucumaria frondosa* and the digestive tract and gonads of the sea urchin *Strongylocentrotus droebachiensis* exhibit significant antioxidant activity [33]. The echinoderms *Astropecten irregularis*, *Luidia sarsi*, and *Ophiura albida* possess promising antioxidant activity, probably due to the accumulated phenolic compounds [5].

Echinoderms are also an important source of various antitumor compounds exhibiting high sensitivity and significant activity. Among the northwestern Pacific invertebrates containing bioactive compounds with experimentally confirmed anticancer potential, members of the classes Asteroidea, Holothuroidea, and Ophiuroidea show the most pronounced effect [34]. Patra with coauthors mention the sea cucumbers *Pseudoconus californica*, *Holothuria impatiens*, *Pharia pyramidata*, *Stichopus chloronotus*, *Holothuria leucospilota*, and *Holothuria scabra* whose extracts exhibited cytotoxicity against several tumor cell lines [35]. Echinoderm extracts have antioxidant, immunomodulatory, and anti-tumor activities owing to the presence of several types of bioactive molecules such as triterpene glycosides, saponins, steroid glycosides, gangliosides, anthraquinone derivatives, polyhydroxysterols, and other secondary metabolites [34,35].

## 3. Fatty Acids of Echinoderms: Structures, Biosynthesis, Dietary Input

Lipids play a vital role in each organism and in marine ecosystems in general, both as major sources of metabolic energy and as important structural components of cell membranes [8]. They play a major role in the physiology (e.g., growth, immunity, and buoyancy), reproductive processes of marine animals, biochemistry (e.g., support of levels of metabolites), and maintenance of membrane fluidity under variable ecological conditions [36]. FAs are common components of complex lipids and are ubiquitously distributed in nature. They have a variety of structures, which differ in the chain length, the number, position and configuration of double bonds, the branching of the hydrocarbon chain, and the presence of functional groups. The diversity of FAs in terms of chain length, degree of unsaturation, and position of the double bonds is responsible for the definitive characteristics of lipids among different organisms. The Echinodermata exhibits a variety of FA structures, which include common, unusual, and exotic FAs (Figure 2, Table 1).

Some FAs are formed through *de novo* biosynthesis in echinoderms. The incorporation of ^14^C-labelled acetate into saturated fatty acids (SFAs) and monounsaturated fatty acids (MUFAs) is higher than that into PUFAs. FAs, being biosynthesized or obtained through specific biotransformation of dietary components, are of great interest for chemotaxonomy. Certain FAs are absent or rare in other animals. The FA composition of echinoderms is generally similar to that of other marine invertebrates and fish in the abundance of PUFAs of n-3 and n-6 series. In most cases, their chain length is shorter than 22 carbon atoms.

The FA composition of echinoderms is characterized by the presence of C12–C24 compounds that are frequently unsaturated (Appendix A). Eicosapentaenoic acids 20:5n-3 (EPA), arachidonic acid 20:4n-6 (ARA), and eicosaenoic acid 20:1n-13 have been identified in many species. Significant differences are found between animals with different feeding modes. An elevated percentage of branched-chain and odd-numbered fatty acids (OBFA) is observed in detritivorous species. This suggests that conversion of dietary FAs is a major source of FAs in echinoderms. The biosynthetic capabilities of echinoderms and the specifics of their diet lead to the emergence and accumulation of a number of unusual FAs. Certain common and unusual FAs or FA groups may serve as chemotaxonomic markers of the classes within the phylum Echinodermata.

FA studies on marine organisms focus mainly on certain PUFAs, which are essential for a multitude of physiological functions in animals. PUFAs are the most abundant categories of FAs in echinoderm lipids, with their content ranging from 20 to 60% of total FA. Among the n-3 family of PUFAs, 20:5n-3 and 22:6n-3 are often found in marine animals and are especially important in maintaining the vital functions in animals and humans (since they also show beneficial effect on human health). The acid 20:4n-6 is a member of n-6, another physiologically important family of PUFAs. Acids 20:5n-3 and 20:4n-6 dominate all classes of echinoderms, although the ratio of the contents of these components varies between classes. While the content of 20:5n-3 prevails over that of 20:4n-6 in Ophiuroidea, the opposite pattern is observed in Holothuroidea and Echinoidea. Another important component of the n-3 family, 22:6n-3, although not a major component in echinoderm lipids, reaches a significant percentage in some members of this phylum (up to 10% of total FAs).

Besides common PUFAs, very long-chain FAs (VLCFA, FAs with more than 22 carbon atoms) such as tetracosahexaenoic or nisinic acid (24:6n-3, THA) are also characteristic of some classes of echinoderms. This acid was first discovered by Takagi with colleagues in Cridoidea and Ophiuroidea, where its content varies from 5 to 16% of total FA [47]. It is also an especially important component in lipids of the Cnidaria species such as jellyfish [48] and soft corals [49]; as a minor component, it has been found in some fish species (Baltic herring, flathead flounder, and eel). The role of THA in the docosahexaenoic acid (DHA) metabolism in the Sprecher pathway is now well documented. It is recognized as a product and precursor to 22:6n-3.

It is probably formed by the following sequence: 20:5n-3 → elongation to 22:5n-3 → elongation to 24:5n-3 followed by further desaturation by Δ6 desaturase [37]. This pathway is assumed for the biosynthesis of 22:6n-3 that is formed by the subsequent chain shortening of 24:6n-3 [50]. In ophiuroids, the last step of the chain shortening does not occur, which results in the accumulation of 24:6n-3. There is evidence that Δ6 desaturase catalyzes the conversion of 24:5n-3 to 24:6n-3 [51]. The relationship between the THA and DHA metabolisms has been proven in vivo on rats: DHA has been demonstrated to be both a product and a precursor to THA [52]. An alternative origin of 24:6n-3 can also be considered, where a high concentration of 24:6n-3 may arise via the elongation of dietary 22:6n-3 [53]. However, this pathway seems unlikely, since diatoms, being the main food of brittle stars, contain a low percentage of 22:6n-3. Nevertheless, it is 20:5n-3 from diatoms that is the precursor of 24:6n-3, since it can be modified by chain elongation and desaturation, leading to the formation of long-chain PUFAs such as 24:6n-3.

A series of MUFAs has been identified in some members of the Echinodermata. Among them, the odd-chain acid 23:1n-9 and the rare isomer 20:1n-13 are of particular interest (Figure 2**)**. It is noteworthy that the distribution of these MUFAs depends on the taxonomic position of the species. The acid 23:1n-9 occurs almost only in the Holothuroidea (1.6–18% of total FAs), and, therefore, can be considered an important chemotaxonomic marker of this class. The acid 20:1n-13 is prominent in lipids of the Ophiuroidea (5–24%) and Asteroidea (15–20%), and it is not found in the Crinoidea. Thus, the combination of the unusual PUFA 24:6n-3 and 20:1n-13 may be a good biomarker of ophiuroids.

Among MUFAs, dominance of the n-9 and n-11 isomers is typical for marine animals, while a high level of the rarely reported isomer 20:1n-13 has been recorded from the coastal brittle star *Amphiura elandiformis* (8.7%) [37]. A mass spectrometric analysis of the dimethyldisulphide (DMDS) adducts for identification of the positions of double bonds in MUFAs have identified five isomers of 20:1 with double bond at Δ5, Δ7, Δ9, Δ 11, and Δ13 positions. Among them, 20:1n-13 (i.e., Δ7) is the major isomer. Mansour with coauthors [37] suppose that in brittle star it may be formed by a specific pathway: by Δ9 desaturase acting on 22:0 with subsequent shortening of formed 22:1n-13, as suggested earlier for fish [39]. It is noteworthy, that the n-13 isomer of 20:1 co-elutes with the n-9 isomer on many GC columns. Thus, the reported occurrence of the 20:1n-9 isomer and the lack of 20:1n-13 in the analyses of brittle stars [53,54] may be in error, since these identifications are based exclusively on GC retention times. It seems possible that 20:1n-9, reported in other papers on brittle stars and other echinoderms, may be mainly the 20:1n-13 isomer.

Numerous findings indicate that 23:1n-9, found almost exclusively in holothurians, is not derived from their diet and is synthesized in their organisms. Kaneniwa, with coauthors, suggested two alternative origins of this FA in animals [38]. First, 23:1n-9 is most likely formed from 24:1n-9 by α-oxidation rather than by desaturation of odd-chain SFAs. In turn, 24:1 is synthesized from n-9 MUFA precursors, including 18:1, 20:1, and 22:1. MUFAs n-9 such as 16:1n-9, 18:1n-9, 20:1n-9, and 22:1n-9 are common components of lipids of holothurians [2,38,55] and many other marine animals. Therefore, the formation of the 23:1n-9 acid may most likely occur through the elongation of precursors by the following biosynthetic pathway with subsequent α-oxidation: Δ9-18:1 → Δ11-20:1 → Δ13-22:1 → Δ15-24:1→ (α-oxidation) 23:1n-9.

The Echinodermata, especially deep-sea species, have proven to be a source of several exotic FAs. The most noteworthy FAs in the mixture are a series of 2-hydroxylated FAs recorded previously from the sea urchin *Tripneustes esculentus* [46] and the sea cucumber *Holothuria mexicana* [45] and later found in abyssal holothurians [2]. Various 2-hydroxy FAs, αOH 22:0, αOH 22:1, αOH 23:0, αOH 23:1, αOH 24:0, and αOH 24:1, were identified in this sea urchin [49]. Furthermore, αOH 18:0 was found in *H. mexicana* [45]. However, while these are minor components (from 0.1 to 1.2% of total FAs) in sea urchins and holothurians from shallow waters, all four holothurian species from the abyssal zone contain 2-hydroxy FAs at extraordinarily high levels: 1.5–4.3% of 2-hydroxytricosenoic acid (αOH 23:1) and 10.5–14% of 2-hydroxytetracosenoic acid (αOH 24:1) [2].

These novel hydroxylated FAs could have arisen from glucosylceramides since they are not found in common phospholipids. As to their biogenesis, it is a matter of speculation at present. These 2-hydroxy FAs could originate from a MUFA oxygenase, possibly lipoxygenase, that converts free FAs into hydroxy FAs. In fact, a calcium-stimulated lipid peroxidizing system has been identified in eggs of the sea urchin *Strongylocentrotus purpuratus* [56].

Another assumption is that they originated as integral parts of the FA chain elongation and/or shortening of bacterial FAs originating from diet by peroxisomes [46]. Moreover, numerous reports have been published on the occurrence of OH FAs in bacteria, especially in those belonging to the family *Flavobacteriaceae*, where these FAs are present as structural components of membranes of phospholipids and lipopolysaccharides [57]. However, bacterial OH FAs are short-chain and have mainly an odd number of carbon atoms and a chain length of C12–C17. The findings of long-chain OH FAs in holothurians indicate the wider than hitherto realized distribution of these components. It should be noted that these components were not detected in the ophiuroids from the same sampling site. It may be interpreted as the ability of holothurians to accumulate these compounds or their precursors from the environment, in particular from microbes that these deposit feeders have consumed. An alternative suggestion on the origin of these FAs can be internal bacteria settled in the gut. Indeed, the Bacteroidetes have been shown to be a dominant phylum with a relative content of 35% in the foregut of *Holothuria atra*. Furthermore, the bacterial community composition in the foregut and hindgut of two common tropical sea cucumbers (*Holothuria atra* and *H. leucospilota*) differs from that recorded from the ambient sediment surface [58].

Non-methylene interrupted (NMI) FAs are detected in the echinoderm classes [9,41,59,60]. NMI FAs are found in a vast variety of marine animals and, therefore, can be considered as common FAs [61,62,63]. However, these are different from usual FAs and refer to FAs with a series of double bonds in which at least one adjacent pair of double bonds is separated by at least two carbon atoms, i.e., by a group other than a single methylene group. In addition to 20:2Δ5,11 and 20:2Δ5,13 and their homologues 22:2Δ7,13 and 22:2Δ7,15, which are typical of marine invertebrates, echinoderms contain also unusual cis-5-olefinic compounds, e.g., 20:3Δ5,11,14, and 20:4Δ5, 11,14,17, which were first identified in polar lipids of 12 sea urchin species from Japan and Canada [44]. Of particular interest may be sciadonic acid, cis-5,11,14-eicosatrienoic or 20:3Δ5,11,14, as an acid exhibiting various biological activities (Section 9). Members of the classes Astroidea and Echinoidea are particularly rich in NMI FAs of different structures. The series of other echinoderm classes arranged in the order of decrease in cis-5-olefinic acids content is as follows: sea urchins, starfishes, ophiuroids, sea cucumbers, and sea lilies.

A characteristic feature of the FA composition of the Echinodermata is the presence of rare or exotic PUFAs, especially in deep-sea species. A number of very long-chain PUFAs, which may be of particular interest, have recently been identified in echinoderms from the abyssal and bathyal zones. Among the common FAs, of which 20:4n-6 and 20:5n-3 are the major ones, a series of homologues 21:4n-7, 22:4n-8, 23:4n-9, and 22:5n-5 has been detected in the sea star *Eremicaster vicinus* and the sea urchin *Kamptosoma abyssale* inhabiting the Kuril–Kamchatka Trench at depths of 5,210–6,183 m [43]. Besides these FAs, a new representative of the n-2 family, 22:6n-2, has been identified in this sea star and this sea urchin (1.6 and 0.3%, respectively) [43]. Since these FAs were found in abyssal foraminifera, the authors suggest that the most plausible explanation for this composition is the presence of foraminifera in the diet of the species considered. It is noteworthy that the rare acid 21:4n-7 has been recorded at a concentration from 0.7 to 6.5% of total FAs from six species of deep-sea holothurians collected in the waters off the Kuril Islands, Sea of Okhotsk (from depths of 90–560 m) [42].

In addition to common FAs with chain lengths between C14 and C24, PUFAs with a chain length of C26 were detected in lipids of five brittle star species collected from the upper bathyal zones off the Kuril Islands, Northwest Pacific, at a depth 340–460 m [43]. These C26 acids, identified as 26:7n-3, 26:6n-3, 26:6n-6, and 26:5n-3, are localized mainly in polar lipids. The concentrations of these acids vary from 0.3 to 4.5% of total FAs, with the major FA being 26:7n-3. According to the authors, the presence of the possible biosynthesis precursors suggests that the C26 PUFAs are produced by brittle stars, and are not accumulated from food sources.

The specifics of FA biosynthesis and composition in different taxonomic groups provide fertile ground for their wide application as chemotaxonomic and biochemical markers of trophic and metabolic interactions in aquatic ecosystems [8]. Nevertheless, the FA profiles of animals are not constant. Diet, growth, reproductive stages, and environmental conditions may exert substantial influence on FA profiles [62]. The factors that affect FA composition are endogenous, exogenous, or both acting in concert. The endogenous factors are associated with the animal growth, life cycle, and annual reproductive cycle and are usually controlled genetically; the exogenous factors are both environmental (temperature, salinity, depth, etc.) and dietary (diet composition, feeding habits, food availability, etc.) [62,64,65,66]. However, it is difficult to attribute inter- and intraspecies variation in FAs to any specific environmental factor, exclusively diet or environmental conditions, because of their mutual interdependence.

Diet exerts a strong influence on FA profiles of animals [8,62,64,65,66]. Most animals cannot synthesize PUFAs from shorter-chain dietary precursors and, therefore, are dependent on preformed dietary PUFAs for maintaining normal physiological functions. FA compositions of most aquatic animals are influenced by FAs of dietary origin. Some of FAs characteristic of primary producers (algae, protists, bacteria, and other microorganisms) cannot be synthesized by animals. Such FAs are found in invertebrates that have consumed certain foods [64,67]. In particular, the sum of *iso*- and *anteiso*-branched chain FAs, 15:0 and 17:0, are used to trace the presence of bacteria in the diet [68,69,70], while consumption of diatoms is indicated by increased proportions of 20:5n-3 and by a high ratio of 16:1n-7/16:0 and C16PUFAs [71,72]. An increased level of 18:4n-3 and 18:3n-4 is the evidence of feeding on dinoflagellates. The prevalence of 18:1n-9 over 18:1n-7 and the abundance of 22:6n-3 suggest the carnivorous mode of feeding [54,62]. Recognizing the significant effect of the food consumed on the FA composition of animals is important for understanding the FA profiles in the echinoderm classes. 

## 4. Fatty Acids of Crinoidea (Sea Lilies and Feather Stars)

Marine animals belonging to the class Crinoidea are the most primitive form of present-day echinoderms. This group flourished in the Paleozoic and Mesozoic and some of its members have survived until now. The approximately 700 crinoid species that have been identified to date typically exist in two forms: unstalked and stalked crinoids. The crinoids attached to the substrate on the seabed with their stalks are commonly referred to as sea lilies because of their resemblance to a flower. These crinoids (comprising about 100 species) are found mainly at depths from 200 to 9,700 m. Unlike sessile sea lilies, the unstalked free-swimming forms, referred to as feather stars (comprising about 600 species), inhabit coral reefs from the intertidal to the deep-sea ocean zones. Crinoids are filter-feeders, consuming primarily microorganisms such as microalgae, mainly diatoms, larvae, and also marine detritus.

Among the studied species, *Tropiometra afra macrodiscus* and *Comanthus japonica* were collected near the coast of Miura Peninsula, Japan [47], and *Heliometra glacialis* was collected from the northwestern continental slope of the Sea of Japan at a depth of 530 m [73] and from the shelves of northeastern Greenland [54]. Although these species differ in habitat conditions, they show rather similar FA compositions with a high content of PUFAs, of which 20:5n-3 (37–25%) and 20:4n-6 (4–17.3%) are the major ones (Figure 3, Appendix A). The FA profiles of crinoids demonstrate a substantial content of nisinic acid, 24:6n-3 (5–10%), which has not been reported as a noticeable component for animals, except for corals [49]. It has been found as a minor component reaching levels below 2% in some of marine fishes. The fact that 24:6n-3 is concentrated in phospholipids of crinoids, rather than neutral lipids similar to 20:4n-6 and 20:5n-3, gives reason to assume 24:6n-3 to be important of the cell functions and structure rather than for storage [47]. The other PUFAs in these three species, such as 16:3n-3, 18:2n-6, 18:3n-6, 18:3n-3, 18:4n-3, 20:2n-9, 20:2n-6, 20:3n-6, 20:3n-3, 21:5n-3, and 22:5n-3 constitute less than 2.5% of total FAs.

The 5-olefinic acids (18:1Δ5, 20:1Δ5, 20:2Δ5,11 and 20:2Δ5,13) found previously in sea urchins [44] were described from sea lilies (at 0.2–1.3%) by Takagi with coauthors [47]. The odd-chain MUFA 23:1n-9, found among the FAs of holothurians at significant concentrations (1.3–17.6%) [2,38,75,76], was detected in Crinoidea lipids only in trace amounts (0.2–0.5%) [47].

The deep-sea stalked crinoid *Dumetocrinus antarcticus* living in the Weddell Sea at a depth of 1,500 m rather differs in the FA composition from the earlier described crinoids [74]. The groups of SFAs, MUFAs, and PUFAs, are quite balanced in this sessile crinoid. Among PUFAs, 20:4n-6 is the most prominent acid (about 17%), 22:6n-3 varies within a range of 3–6%, 20:5n-3 does not exceed 4%, whereas the acid 16:0 is abundant (about 18%). This species exhibits various MUFAs with chain lengths from C16 to C24. All the crinoid species contain the acids 20:1n-9 and 22:1n-11 characteristic of zooplankton, which indicates their dietary origin. The authors did not identify 24:6n-3 in *D. antarcticus* [74] but detected an unusual acid, 24:5n-3. The retention times of these long-chain PUFAs are very similar and, thus, the reported occurrence of 24:5n-3 at level of about 2% and the lack of 24:6n-3 in the analysis of this crinoid species [74] may be an error due to the identification based solely on GC retention times.

Crinoids are filter-feeders capturing microscopic organisms such as foraminifera, larvae, microalgae, small crustaceans, and radiolarians, as well as organic detritus. The differences in the diet of these animals lead to variations in the FA composition between them. Kharlamenko with coauthors emphasized the effect of diet on the FA composition of crinoids [73]. Lipids of the deep-sea *Heliometra glacialis* contain both FAs typical for diatoms such as 16:4n-1 and 20:5n-6 and FAs characteristic of zooplankton such as 20:1n-9 and 22:1n-11. Therefore, this species feeds mainly on zooplankton inhabiting the euphotic zone and additionally ingests phytoplankton.

## 5. Fatty Acids of Ophiuroidae (Brittle Stars, Serpent Stars, or Ophiuroids)

Ophiuroids, often referred to as brittle stars or serpent stars, comprise over 2000 species and make up the largest group of echinoderms. They generally have five long, slender, serpent-like arms, which they use for locomotion. Unlike starfish, brittle stars do not rely on tube feet to move. With their arms they can walk and swim freely and quite fast. These organisms are widely distributed across the world’s oceans. Being often keystone species in soft sediment communities, ophiuroids are more frequently found in deep-sea waters, at depths greater than 200 m. They can, however, also inhabit coastal areas and reefs, hiding deep in corals or under rocks. Although these species are of little commercial value, they play an important role in many benthic food webs. Ophiuroids are generally omnivorous macro- or microphages. According to their lifestyle, their feeding modes can be suspension feeding, deposit feeding, and predation; some species use more than one mode to obtain food.

An analysis of the available literature has shown that ophiuroids’ FAs are quite extensively studied. Among the reported species are inhabitants of temperate, tropical, Arctic waters, and also species from the intertidal zone of coastal waters (to 40 m) [37] and the deep-sea zone [2] with depths greater than 4000 m). Some of them have symbionts, while others do not [53]. Although the percentages of the major FAs are quite variable in brittle stars, the FA compositions of the studied ophiuroids show a number of generally common patterns (Appendix A). The dominant FAs in these species are 20:5n-3, 20:1n-13, and 24:6n-4, with a low level of 22:6n-3 regardless of locality, environmental conditions, and sampling date (Figure 4). The major PUFA, 20:5n-3, ranges from 7.1% in *Ophiozonella longispina* [77] to 32.8% in *Ophiacantha bidentata* [54]. The percentage of 20:4n-6, which is abundant in various echinoderm species, varies in ophiuroids from 1.2% in *Ophiopholis aculeata* [54] to 8.3% in *Macrophiothrix longipeda* [77], and only in *Ophiura carinifera* it is the most prominent FA (about 30%) [74]. The authors suggest that it is derived from omnivorous diet. The SFAs 14:0, 16:0, and 18:0, as well as 18:1, occur in moderate amounts. The tetracosahexaenoic acid 24:6n-3 and the isomer 20:1n-13 are reported to be present in ophiuroids at high percentages. Thus, the combination of these rare FAs may be a good biomarker of brittle stars in food web studies. Earlier, Mansour, with coauthors, proposed 24:6n-3 as a biomarker for brittle stars [37].

The rarely reported isomer 20:1n-13 has been found in the coastal brittle star *Amphiura elandiformis* in large amounts (8.7%) [40]. Similar values of 20:1n-13 were recorded from the brittle star *Ophiura leptoctenia* collected from the Sea of Japan continental slope (8.5%) [73]. However, the microbial food web is assumed as another more probable pathway of 20:1n-13 in brittle stars. According to the authors, the high 20:1n-13 content indicates that microbial food web makes a certain contribution to the diet of this species that ingests bacteria along with zooplankton [73]. Moreover, 20:1n-13 was found to be present in comparable concentrations in some invertebrates including another echinoderm, the sea star *Ctenodiscus crispatus* [73]. It is noteworthy that the n-13 isomer of 20:1 is not always identified in brittle stars, because it co-elutes with the n-11 and n-9 isomers on many GC columns, and the retention times of these isomers are similar. Before Mansur with colleagues [37] accurately identified and proved the dominance of the isomer 20:1n-13 in brittle stars, the main isomers had been reported to be n-9 [53,54], n-11 [2,47], or sum of isomers [41,79], or some authors had not determined the positions of double bonds in 20:1 [77,78]. Thus, the reported dominance of the 20:1n-9 or 20:1n-11 isomers and the lack of 20:1n-13 in brittle stars may be an error since these identifications were based solely on GC retention times.

Among MUFAs, the unusual long-chain acid 23:1n-9 was found first in lipids of the ophiuroids *Asteronyx loveni* (0.23%) and *Ophioplocus japonicus* (0.36%) [47]. Then this acid was identified only in three other brittle star species as a minor component (0.4–0.7%): *Ophiura sarsi* [41], *Ophiura bathybia*, and *Ophiacantha* sp. [2].

The structure of the unusual long-chain PUFA, the nisinic acid, Δ6,9,12,15,18,21-tetracosahexaenoic acid, 24:6n-3 (THA), was originally determined in two species of Ophiuroidea, *Asteronyx loveni* and *Ophioplocus japonicus*, in which it is present in marked amounts (4.6% and 7.9%, respectively) [47]. It is detected also in other brittle star genera species regardless of their habitat conditions [37,53,73]. Its content ranges from 2.6% in *Macrophiothrix longipeda* [77] to 15.5% in *Ophiocomina nigra* [53]. A large amount of this FA was found in *Ophiura sarsii* (10.8–13.9%) [77,78]. It is noteworthy that the sea cucumber *Stichopus japonicus* collected in the same area [47] contained no THA despite the fact that this animal also feeds on detritus [38,75]. Therefore, these results do not confirm a dietary source for this FA. Other echinoderms such as sea stars, sea urchins, and sea cucumbers seem to lack 24:6n-3.

Although the physiological role of THA in the organism is still poorly understood, it was already reported that THA in echinoderms is concentrated in phospholipids rather than in triacylglycerols [47]. A comparison of THA content between the lipid classes in the brittle star *Ophiura sarsi* shows that the THA in phosphatidylcholine and phosphatidylethanolamine makes up around 20%, while in triacylglycerol it is less than 10% [78]. The dominance of this PUFA in polar lipids, in particular in phosphatidylcholine and phosphatidylethanolamine, suggests its structural and functional role in the organism.

Structural analyses of FAs of *Ophiura sarsi* collected at a depth of 1,100 m have revealed a number of extraordinary FAs belonging to a family of NMI PUFAs with mixed geometry of ethylenic bonds that were not reported previously [40]. The structures of the unidentified FAs were determined as 7*E*,13*E*-eicosadienoic (20:2) (2.2%), 7*E*,13*E*,17*Z*-eicosatrienoic (20:3) (12.7%), 9*E*,15*E*,19*Z*-docosatrienoic (22:3) (5.6%), and 4*Z*,9*E*,15*E*,19*Z*-docosatetraenoic (22:4) (3.0%) acids by GC-MS of dimethyl disulfide adducts and GLC of monoenoates on a polar column. Neither before nor after this report was any information ever published about such extraordinary NMI FAs in brittle stars.

Another unusual FA, C26 PUFA, was found in *Ophiacantha* sp. and in *Ophiura bathybia* from habitats located at 4,100 m in the Northeast Pacific. In these species, it makes up 11 and 1%, respectively [2]. However, the authors do not specify the double bonds positions in this compound.

Taking into account that ophiuroids live on soft bottom sediments and are mostly deposit feeders, it is not surprising that they contain an increased level of bacteria-specific OBFA, mainly due to *iso*- and *anteiso*-15:0, 15:0, 15:1, *iso*- and *anteiso*-17:0, 17:0, and 17:1. The total OBFA content varies from 0.8% in *Ophiothrix fragilis* [53] to 5.0% in *Amphiura elandiformis* [37] and even to 7.5% in *Ophiura sarsi* [41]. The major source of OBFAs in brittle stars is bottom sediment bacteria that constitute a substantial part of the diet of these benthic invertebrates.

In addition to common FAs with chain lengths between C14 and C24, several unusual C26 PUFAs were also found in five deep-sea species of Ophiuroidea [43]. The long-chain FAs 26:7n-3, 26:6n-3, 26:6n-6, and 26:5n-3 were identified by hydrogenation, GC-MS of the methyl esters, and 4,4-dimethyloxazoline (DMOX) derivatives. Their concentration varies from 0.3 to 4.5% of total FAs. The authors suggested that the C26 PUFAs are produced by brittle stars, and are not derived from food sources.

Brittle stars are generally known to be very opportunistic in their feeding modes and food sources. According to their life styles, they have evolved different feeding modes such as suspension feeding, deposit feeding, scavenging, predation on live organisms, and even cannibalism [80]. Previous studies considered ophiuroids as generally omnivorous macro- or microphageous feeders [81]. The observed interspecific differences in their FA composition are explained by the diversity of food sources of these animals. Many of the FAs in brittle stars are primarily derived from their diet that obviously includes microalgae, animals, bacteria, and detritus.

The differences in the FA profiles between eight species are clearly visualized by principal component analysis (PCA) (Figure 5). The FAs that contributed most to the separation of the groups along PC1 are 20:1, 20:5n-3, C18PUFA, and 18:1n-9, which collectively explain 78.3% of the total variance. The species separated along PC2 explain 11.8% of the total variance. The major FAs contributing to PC2 are 20:1, 20:5n-3, 20:0, and 14:0. It is noteworthy that three specimens of carnivorous *Ophiurs sarsii*, although collected from different localities [49,77], formed a single group. This suggests that the dietary differences between the ophiuran species are much more pronounced than the intraspecific FA differences for individuals of these species from different habitats. Figure 5B shows projections of 24 variables significant for the separation of ophiuroid species. Group 1 consisting of 20:5n-3, 16:1n-7 and 14:0 combines FA markers of diatoms [71,72]. The presence of these FAs in large quantities indicates an important contribution of diatoms in the diet of brittle stars. Group 2 includes 18:1n-9, 22:6n-3 and the product of its elongation 24:6n-3; these FAs are indicators of carnivorous feeding mode [66,82]. Group 3, which can be clearly distinguished, consists of OBFAs, C20 and C22 NMID FAs, 18:1n-7, and hydroxy FAs (OH-23:1 and OH-24:1). This group combines FAs of bacterial origin [57,68,83]. Thus, the FA biomarker approach confirms the observation on the feeding modes of brittle stars: these are mostly suspension feeders, deposit feeders, and predators.

The spatio-temporal variations in FA profiles between two common co-occurring ophiuroids, *Ophiocomina nigra* and *Ophiothrix fragilis*, are generally low, in contrast to interspecific differences being more pronounced. The FA markers have shown that ophiuroids rely mainly on diatom inputs. In addition, the low PUFA/SFA ratio, along with the elevated level of bacterial FA markers, indicates that *O. nigra* consumes mostly detritus, while *O. fragilis* prefers fresh phytoplankton-derived material [84].

The effect of diet on the distribution of animal FAs is confirmed by a comparison of four Arctic species of brittle stars from the shelves of northeast Greenland and the Barents Sea [54]. The amounts of the major FAs in them are quite variable. Both *Ophiopleura borealis* and *Ophiura sarsi*, reported to prefer predatory feeding, showed comparatively high levels of 22:6n-3 (5 and 8.5%, respectively) and a high percentage of 18:1n-9 in contrast to the level of its isomer, 18:1n-7 [54]. Usually, a high 18:1n-9/18:1n-7 ratio and a high level of 22:6n-3 reflect the carnivorous feeding mode and indicate that the animals have fed on zooplankton from the water column. *Ophiacantha bidentata* had the largest amount of 20:5n-3 (32.8%), a ratio 16:1n-7/16:0 equal to 1.1, and a very low level of 18:1n-9 suggesting that it consumed mainly diatoms. *Ophiopholis aculeata* from the shallow-water Spitsbergen Shelf station was particularly rich in 18:4n-3 (24.4%) [54]. This FA indicates a phytoplankton food source since it is abundant in dinoflagellates [71,72].

Moreover, a distinct intraspecific difference was observed in *Ophiopleura borealis* from the Svalbard shelf as compared to samples from the northeastern Greenland shelf [54]. The Svalbard samples have high proportions of 20:1n-9 (22%). Taking into consideration that the n-11 isomer of 20:1 may co-elute with the n-9 isomer on many GC columns, it seems possible that 20:1n-9 reported in this paper may be mainly the 20:1n-11, which is a marker of zooplankton in diets of animals. In the animals from the Greenland shelf, this FA occurs only in low proportions but its FA composition is dominated by 20:5n-3 and 16:1n-7, which are considered the markers of diatom input [62,65,71].

The significant differences in relative proportions of FAs between *Ophiothrix fragilis*, *Amphiura chiajei*, and *Ophiocomina nigra* from Oban Bay, Scotland, result from their feeding habits and symbiosis with chemoautotrophic bacteria [53]. *Ophiothrix fragilis* is exclusively a suspension feeder, while *A. chiajei* is a deposit feeder with opportunistic scavenging tendencies. *Ophiocomina nigra* is an active carnivore and scavenger. The high levels of 16:0, 16:4, 18:4n-3, and 20:5n-3 in *O. fragilis* suggest a diet rich in phytoplankton, particularly diatoms and dinoflagellates [71,72]. The elevated level of 20:1n-9 in *A. chiajei* and *O. nigra* indicates ingestion of copepods. The high concentration of 16:1n-7 in symbiotic species *O. fragilis* and *A. chiajei* compared to that in the non-symbiotic *O. nigra* and the elevated level of 18:1n-7 in the symbiotic *O. fragilis* are indicators of bacterial input in the host’s diet, as the authors state [53].

This opinion is supported by the data of a FA study on *Amphiura elandiformis* [37]. It is a very common species that inhabits the soft bottoms off of southern Tasmania, Australia, where it burrows into the sediment, leaving its arms above the surface to capture plankton in the flowing water. Indeed, the dominance of 14:0, 16:1, 16:0, 20:4n-6, 20:5n-3, and a low amount of 22:6n-3 in this species is mainly due to the consumption of diatoms. Among four species from off Newfoundland, eastern Canada, Northwest Atlantic, *Ophiopholis aculeata* and *Gorgonocephalus* sp. exhibit elevated amounts of 20:5n-3, a high 16:1n-7/16:0 ratio, and noticeable amounts of C16PUFAs such as 16:2n-4, 16:3n-4, and 16:4n-1 [9], thus, indicating the importance of diatoms in their diet. *Ophiacantha* sp. from the abyssal zone [2] exhibits a low percentage of 24:6n-3 (about 2%), which is a derivative from dietary 20:5n-3, and an unusually high level of 20:0 (11.8%), which is commonly detected in trace amounts. However, another species from the same zone, *Ophiura bathybia*, has not shown such features. The authors suggest that ophiuroids’ FA compositions indicate the consumption of animal material in addition to phytodetritus.

One of the explanations for the discrepancy of the results with regard to the FA variability may also be methodological inconsistencies. This is especially clearly evident in the case of FAs, such as the widely distributed species as *Ophiura sarsi*, where fluctuations in levels of some FAs are extraordinary. The 24:6-3 content showed pronounced differences from 13.9% [78] to complete absence [54,79]. Sato with coauthors reported results on this species that are fundamentally different from those published by other authors. The authors identified extraordinary polyenic NMI FAs with *cis*- and *trans*- double bonds at high contents (with a total of 22.3%), which had never been found before. Moreover, in *Ophiura carinifera*, 24:6n-3 has not been detected, but 24:4 and 24:5n-5 have been identified (about 4%) [74].

## 6. Fatty Acids of Asteroidea (Sea Stars, Starfish)

The members of the class Asteroidea are referred to as sea stars, or starfish, because of their star-shaped bodies. To date, approximately 1900 species of these marine benthic invertebrates are known. They inhabit the seabed in all the world’s oceans, from the tropics to cold polar waters and from the intertidal to abyssal zones, to a depth of 6000 m. Sea stars are slow moving animals that slide over the bottom on the tips of their tube-feet. Their arms, or rays, may be very short and broad, radiating from the large disk; otherwise, they may be very long and rather narrow, with the body being quite small. The number of arms is commonly five, but members of some genera (*Solaster*, *Brisinga*) have more arms, 9 to 15.

These are opportunistic feeders and are mostly predators preying on benthic invertebrates. Several species have specialized feeding behaviors including eversion of their stomachs and suspension feeding. Asteroids are known to exhibit intraspecific feeding diversity. Most are generalist predators, consuming microalgae, sponges, bivalves, snails, and other small animals. Some are monofagous, such as the crown-of-thorns starfish, *Acanthaster planci*, that preys on coral polyps, while others are detritivores. The *Henricia* species are suspension feeders, filtering phytoplankton. Several species are known to absorb nutrients from the sea water. Sea stars display a range of feeding modes, from predation to detritus-feeding. Most of them prey on various animals, mainly echinoderms, mollusks, and worms. Some are particulate suspension- and non-selective deposit-feeders living in temporary burrows that they make in the upper sediment layer.

A vast variety of species have been studied, including those from temperate and cold waters, coastal, deep-sea habitats, and inhabitants of the abyssal zone. The FA profiles of asteroid species are certainly different but show some evident general similarities (Appendix A). The most prominent FAs are 20:5n-3, 20:4n-6, and 20:1. Among them, the most abundant FAs across species are 20:5n-3 reaching 32.5% in *Crossaster papposus* [79], 20:4n-6 (29.4%) in *Myxaster sol* [9], and 20:1 (31.3%) in *Zoroaster longicauda* [60]. In some of the species, the dominant FA is 20:5n-3, while in others, it is 20:4n-6, with the 20:5n-5 to 20:4n-6 ratio ranging from 0.2 in *Mediaster bairdi* [9] to 4.4 in *Crossaster papposus* [79] that is particularly rich in 20:5n-3 (32.5%). The acid 20:4n-6, with its content in most species exceeding 20% of total FAs, plays a major role in sea stars. It is particularly abundant in the deep-sea *Ctenodiscus crispatus* (32.1%), in *Myxaster sol* (29.4%) [9], and even in *Ctenodiscus crispatus* (32.1%) [73], while an extremely low level of 20:4n-6 was recorded from *Brisingida* sp. (0–1.45%) [9]. Similar to most members of the Echinodermata, sea stars have negligible amounts of 22:6n-3, except for *Asterias amurensis* (in which this FA reaches 10.9%) [77].

The dominant MUFA in all the reported species is 20:1, being more abundant in *Zoroaster longicauda* (31.3%) [60]. In the cases where the n-13 and n-9 isomers are identified separately, the n-13 isomer dominated (13–19%) [60]. Among the 20:1 isomers identified in the species collected from the coastal water off Hakodate, Japan, 20:1n-11 is a major acid (4.7–12.5% of total FAs) [59]. It is apparently the 20:1n-13 isomer. Another important MUFA is 18:1 with the major isomer being 18:1n-7. 

Odd-chain and branched FAs found in sea stars reached in total 5–7% in some species such as *Myxaster sol* (5.9%), *Mediaster bairdi bairdi* (5.7%) [9], and *Distolasterias nippon* (7.5%) [85]. Large amounts of branched FAs (up to 13.5%) were detected in *Asterias amurensis* from mud bottoms, mainly due to *anteiso*-15:0 and *anteiso* 17:0 [59]. The OBFAs typical of bacteria in sea stars certainly originate from bacteria of bottom sediments directly or through the food web. The highest percentages of bacterial FAs have been recorded from the abyssal species *Eremicaster vicinus* (17.4%) that consumes bacteria along with deep-sea foraminifers, which, in turn, feed on bacteria [41].

NMID FAs, frequently detected in marine organisms, are not always found in sea stars. It is probably due to the lack of attention to these minor components. The content of NMID FAs in sea stars ranges from 0.4 to 9.5%. The maximum level of these FAs has been recorded from *Hyphaster inermis* [60] and the minimum level from *Mediaster bairdi* [9]. Among them, the isomers 20:2Δ5,11 and 20:2Δ5,13 usually prevail over 22:2Δ7,13 and 22:2Δ7,15. In addition to these dienoic NMID FAs typical of mane marine invertebrates, some unusual NMI PUFAs including all-*cis*-5,11,14-eicosatrienoic acid (20:3 Δ5,11,14) and all-*cis*-5,11,14,17-eicosatetraenoic acid (20:4 Δ5,11,14,17) were also identified in five Asteroidea species collected off Hakodate [59]. The authors concluded that these FAs are synthesized in sea stars and are not incorporated in their organisms from bivalve mollusks, which are their main food. Although C20 NMI FAs dominate in sea stars, these are found only in trace amounts in mollusks.

Even more exotic FAs have been identified as minor components in the abyssal sea star *Eremicaster vicinus* [41]. Among them, such NMID FAs as 20:2Δ7,13 and 20:2Δ7,15 and their homologs 22:2Δ7,15 and 22:2Δ7,15 have been found. Furthermore, a series of homologous unusual PUFAs, 21:4(n-7), 22:4(n-8), 23:4(n-9), and 22:5 (n-5), are reported. The presence of these FAs characteristic of bacteria and deep-sea foraminifera suggests their dietary origin in this abyssal sea star [41].

Although differences between species may partly result from methodological inconsistencies in the FA analysis, the FA profiles of sea stars largely depend on their trophic preferences. This statement is confirmed also by the clustering of nine deep-sea species, which divides them into various trophic groups. The FA profile of these species is clustered on the basis of their feeding modes as suspension feeders, predators/scavengers, and mud ingesters [60].

The PCA has divided 21 species into six groups depending on their FAs (Figure 6). The analyzed sea star species represent different marine ecosystems of the Pacific Ocean [9,77], a cold coastal ecosystem of Gilbert Bay, Labrador [79], and deep-sea waters off Newfoundland [9]. These groups correspond to several feeding modes.

Group 1 comprises *Pteraster* sp. and *Crossaster papposus*, which are apparently predators preying on other echinoderms. The large-sized sunstar *Crossaster papposus* is considered an omnivore that prefers other echinoderms. It also preys on other animals including sea pens, nudibranchs, and bryozoans. Group 2 consists of *Luidia quinaria*, *Asterias amurensis*, *Distolasterias nippon*, *Brisingida* sp., *Solaster paxillatus*, *Leptychaster arcticus*, and *Zoroaster fulgens*. These are predators preying on various benthic invertebrates. Group 3 comprises two species of *Astropecten*, mainly carnivorous, preying on small mollusks. *Astropecten americanus* exhibits a relatively high selectivity as evidenced by stomach contents. It generally ingests small-sized prey, mainly mollusks and also crustaceans. Group 4 comprises *Ctenodiscus crispatus*, *Certonardoa semiregularis*, and *Eremicaster vicinus* that are non-selective deposit feeders. These asteroids, living in temporary burrows in the uppermost sediment layer, are mud-ingesters feeding on small invertebrates, algae, and detritus. Group 5 consists of three species: *Freyella microspina*, which is known as suspension feeder; *Psilaster andromeda*, which prefers a soft, muddy bottom and displays a range of feeding modes, from predation to detritus-feeding. The stomach contents of the third species, *Cheiraster*, indicate omnivorous or predatory feeding behaviors, but suspension feeding has also been suggested. Group 6 comprises *Myxaster sol*, *Mediaster bairdi*, and *Ctenodiscus crispatus*. In the gut of *Mediaster bairdi*, the presence of benthic foraminifera and traces of sediments have been registered. *Ctenodiscus crispatus* are particulate suspension and non-selective deposit feeders living in temporary burrows in the uppermost sediment layer.

## 7. Fatty Acids of Holothuroidea (Sea Cucumbers)

Echinoderms from the class Holothuroidea are marine invertebrates with a leathery skin and a characteristic elongated body for which they are referred to as sea cucumbers. Externally, holothurians look unlike other echinoderms due to their elongated body without visible skeleton and the lack of arms that many other echinoderms such as sea stars and brittle stars have. To date, there are over 1400 known holothurian species in the world. They are inhabitants of the sea bed distributed worldwide. These animals are apparently well adapted to extreme depths, where they display rich species diversity. Holothurians often dominate the benthos and are considered one of the most characteristic animals of the abyssal ocean zone. Sea cucumbers can form dense populations in shallow waters and at abyssal depths.

FA compositions of a large number of sea cucumber species from various regions have been documented in detail (Appendix A). Although the overall FA profiles of the Holothuroidea vary, there are some similarities across the species characterized by high levels of 20:4n-6, 20:5n-3, 23:1n-9, 16:0, 18:1, 22:6n-3, and 16:1n-7, and also by the presence of noticeable amounts of branched FAs, *iso*-15:0, *iso*-17:0, *anteiso*-17:0, in their tissues (Figure 7, Appendix A).

PUFAs dominate total FAs in all the studied holothurians, except for the abyssal species *Oneirophanta mutabilis* (9.2%) [2] and *Holothuria leucospilota* from temperate waters (14.8%) [59]. The PUFA contents range from 29.6% in *Benthogone rosea* [86] to 58.1% in *Deima validium* [76]. The levels of PUFAs in most of the analyzed holothurian species usually account for about a half (50%) of total FAs. The major PUFAs for all species are 20:4n-6 and 20:5n-3, with 22:6n-3 present in much lower proportions. The acid 20:5n-3 reaches especially high levels in FAs of *Cucumaria* sp.: 36.9% for total lipids and 42.7% for polar lipids [38]. The body wall of *Cucumaria* sp. also contains a particularly high proportion of 20:5n-3, 28.4% of total FAs [55]. However, a specific feature for many of holothurians is the substantial prevalence of 20:4n-6 over 20:5n-3. A high 20:4n-6 to 20:5n-3 ratio was recorded from both coastal and abyssal species. For example, the tropical *Holothuria pardalis* contains 22.3% of 20:4n-6 and only 3.5% of 20:5n-3 [75], while the abyssal *Paroriza pallens* contains 30.6% and 8.1%, respectively [86]. Nevertheless, a comparison of the FA compositions of the species from tropical and temperate waters has shown the evident (multifold) prevalence of 20:4n-6 over 20:5n-3 in tropical species [75], whereas the ratio for species from temperate waters is reverse, with the level of 20:5n-3 being substantially higher than 20:4n-6. This becomes particularly evident in the case of *Eupta fraudatrix*, in which the content of 20:5n-3 is 28.7% and that of 20:4n-6 is only 3.1%.

The differences in the contents of these components in species from temperate and tropical latitudes result largely from the differences in their diets. Organic matter rich in microalgae and bacteria is the major food supply for holothurians [73]. The plankton and bottom sediments from Peter the Great Bay, Sea of Japan, is known to be dominated by diatoms that are characterized by abundance of 20:5n-3 and prevalence of 16:1n-7 over 16:0 [71,72]. In the holothurian species containing high levels of 20:5n-3 such as *Stichopus japonicus* (22.4%), *Eupta fraudatrix* (28.7%) [75], *Cucumaria* sp. (36.9%) [38], *Chiridota laevis* (23.3%) [79], and *Molpadia blakei* (25.8%) [76], the 16:1n-7/16:0 ratio is much greater than 1 (1.4, 2.2, 1.3, 1.2, respectively). This confirms the dietary origin of these FAs and that diatoms constitute a significant portion in the diet of these species. In fact, Japanese sea cucumber *Stichopus (=Apostichopus) japonicus* sifts through sediments on the sea bed with its tentacles and feeds on detritus and other organic matter containing plant and animal remains, bacteria, protozoans, and diatoms. The sea cucumber *Cucumaria frondosa* is a benthic suspension feeder that captures food particles on its tentacles and then sends them into its mouth. Its diet comprises an abundance of phytoplanktonic diatoms (*Coscinodiscus centralis*, *Chaetoceros debilis*, *Skeletonema costatum*, and *Thalassiosira gravida*), occasionally small crustaceans, and a variety of eggs and larvae.

An analysis of FA biomarkers has shown that the main food sources of the sea cucumber *Apostichopus japonicus* change with seasons [87,88]. Diatoms, flagellates or protozoans, brown alga and bacteria are the main diet in January. The contributions of diatoms, flagellates or protozoans, and green microalgae (Chlorophyta) are comparatively high in March. Chlorophytes are the most important food in June. The diet of the sea cucumber consists mainly of bacteria and chlorophytes in July. Bacteria are the most important food source of *A. japonicus* in August and September, while brown algae and bacteria constitute a considerable portion of the diet from October to November [87]. The variations in the FA compositions of the three holothurian species from bathyal and abyssal depths are found to coincide in time with the high seasonal fluxes of phytodetrial organic matter to the sea bed in the northeastern Atlantic Ocean. Furthermore, deep-sea holothurians display a high degree of interspecific differences in their FA composition that may be related to the different reproductive strategies [86].

The average content of 22:6n-3 in holothurians is 3–5%. The lowest level is observed in *Chiridota laevis* from coastal bays in Newfoundland (0.1%) [79] and the highest level is in *Amperima rosea* (9.4%) and *Bathyplotes natansfrom* (11%) from the abyssal zone of the Northeast Atlantic [86]. Among all the holothurian species under consideration, the very long chain PUFA 24:6n-3 was detected only in two species and only in trace amounts (0.1–0.2%) [2].

Another feature of holothurian FAs is the presence of an unusual odd-chain MUFA, 23:1n-9 (tricosenoic acid), in significant amounts. Marine animals usually contain even-numbered FAs [8,64]. The structure of this MUFA was determined as *cis*-14-tricosenoic acid 23:1n-9 for the first time in lipids of *Stichopus japonicus*, *Cucumaria* sp., and *Holothuria leucospilota* from shallow coastal waters of Japan [38]. Its content ranged within 1–8% in these species. The highest levels of this acid were reported for the abyssal species *Deima validium* (17.6%), *Psychropotes longicauda* (17.4%), and *Oneirophanta mutabilis* (17.1%) from the northeastern Atlantic Ocean (Porcupine Abyssal Plain) [76]. In *O. mutabilis* from the northeastern Pacific Ocean, collected 220 km west of Point Conception, California (Monterey Abyssal Fan), the level of 23:1n-9 is only 8.8% [2]. However, this FA was not always reported for holothurians including *O. mutabilis* from the abyssal zone [86] and from costal bays [79]. This difference could possibly be due to the co-elution of 23:1 with 20:4n-6 in the study where short packed columns instead of capillary columns were used [2].

Kaneniwa, with coauthors, suggested two alternative origins of this FA in animals: 23:1n-9 is most likely formed from 24:1n-9 by α-oxidation rather than by desaturation of odd-chain SFAs; in turn, 24:1 is synthesized from n-9 MUFA precursors, including 18:1, 20:1, and 22:1 [38]. MUFAs n-9 such as 16:1n-9, 18:1n-9, 20:1n-9, and 22:1n-9 are common components of lipids in holothurians [2,38,55], as well as in many other marine animals. Therefore, the formation of the 23:1n-9 acid may most likely occur through the elongation of precursors by the following biosynthetic pathway with subsequent α-oxidation: Δ9-18:1 → Δ11-20:1 → Δ13-22:1 → Δ15-24:1→ (α-oxidation) 23:1n-9.

The pathway of 23:1n-9 biosynthesis from 24:1n-9 by α-oxidation is described in Section 3. Diet of animals can be considered an alternative source of 23:1n-9. However, the sea urchins collected at the same time and locality as the holothurians lacked 23:1, yet had a similar detrital diet [38]. Similarly, the brittle stars from the same area (Monterey Abyssal Fan) contained only trace amounts of 23:1 (0.6–0.7%) compared to the holothurians (4.2–8.8%) [2]. This finding suggests that 23:1n-9 in holothurians is not derived from their diet and is synthesized in organisms.

The physiological role of this FA is not fully clear. Nevertheless, the amount of 23:1n-9 in the body wall of *Cucumaria okhotensis* and *C. japonica* is significantly higher than in viscera tissues: 13.1% vs. 1.0% and 7.7% vs. 2.2%, respectively [55]. Its concentration is markedly higher in polar lipids compared to neutral lipids, similarly to that observed for the essential acids 20:4n-6 and 20:5n-3 [38], which indicates the importance of 23:1n-9 for the membrane structure and function. 

Holothurians from different habitats show remarkable differences in relative amounts of SFAs such as 14:0, 16:0, 18:0, and 20:0. The average amount of SFA in deep-sea species account for 2.6–6.3% [76] or 6.2–9.4% [2], which is significantly lower than that reported for species from temperate (21.6%) [38] and tropical (23.4%) waters [75]. The elevated levels of 23:1 and other MUFAs in the deep-sea holothurians are suggested to result from the necessity to maintain the membrane fluidity at low temperature and high pressure [76].

Unlike other representatives of echinoderms, holothurians possess a high amount of OBFAs typical of bacteria. These bacterial FA markers indicate a greater microbial input to the diet and assimilation into tissues in holothurians than in other invertebrates [67,70]. Although their percentage varies between species, this pattern is, nevertheless, evident. For example, FAs with odd-number chain and branched FAs are reported for *Stichopus japonicus* (7.5 and 8.9%, respectively), *Cucumaria* sp. (9.9% and 9.9%), and *Holothuria leucospilota* (6.6 and 6.5%) [38]. It is noteworthy that species from temperate waters (Sea of Japan) contain much more OBFAs than species from tropical water (off southern Vietnam) [75]. The OBFA content reaches 18.3% in *Eupentacta fraudatrix* [75], 14.3% in *Cucumaria japonica*, and 12.6% in *C. okhotensis* [55]. Proportions of OBFAs are commonly used to estimate the input of bacteria in the diet of invertebrates [65,89]. Considering that bacteria contain a high level of OBFAs [68,89], it is obvious that these FAs in holothurians originate from bacteria of bottom sediments. Indeed, these animals feed upon the surface layer of bottom sediments, which is rich in organic detritus and bacteria. The significant difference in the level of these components between species from temperate and tropical waters is evidently due to the difference in their diet [75].

However, results of an FA analysis may differ fundamentally for the same species collected from different regions. For example, the black sea cucumber (or black tarzan) *Holothuria leucospilota*, a common species of shallow waters along the eastern coast of Africa, the Indo-Pacific region, and the northeastern coast of Australia, is a scavenger that feeds by using its tentacles to shovel organic debris from the sea bed and send them into its mouth. The FA composition of *H. leucospilota* from southern Vietnam waters shows a dominance of PUFAs (52.3%) with the major 20:4n-6 (27.2%), 20:5n-3 (5.6%), and 22:6n-3 (6.4%), a significant input of 20:1 (13%) and 23:1 (6.7%), and also a low content of OBFAs (2.2%), 16:0 (5.4%), 18:0 (3.8%), and 16:1n-7 (2.5%) [75]. However, this species collected in the Sea of Japan shows the opposite tendency, with an abundance of OBFAs (13.1%), 16:0 (13.6%), 18:0 (16.6%), and 16:1n-7 (11.8%) and a low content of PUFAs (14.8%) composed mainly of 20:4n-6 (5.5%), 20:5n-3 (1.5%), and 22:6n-3 (1.4%) [38]. Undoubtedly, these differences in the FA composition arise from the difference in consumed and available food.

Other holothurian FAs deserve special mentioning. NMI FAs such as 20:2Δ5,11 and 20:2Δ5,13, and their homologies 20:2Δ7,13 and 22:2Δ7,15, known from various taxonomic groups of marine animals [62,63], are specific for holothurians. In contrast to other classes of the phylum Echinodermata, the Holothuroidea contain noticeable amounts of C22 NMID FAs (2.1–14%) and low amounts of C20 NMID (0.1–2.5%) [2,79,86]. It is likely that holothurians obtain these components by the biosynthetic pathway described for marine mollusks [61,83]. The 5-olefinic acids, 18:1Δ5, 20:1Δ5, were identified as minor components (within 0.5–1.1% of total FAs) [38].

In addition to common FAs, several unusual FAs are also found in the Holothuroidea. A substantial amount of the hydroxy acids, αOH 24:1 and αOH 23:1, was detected in four species from the abyssal zone in the eastern North Pacific. The acid αOH 24:1 amount to 10.5–14.1%, while αOH 23:1 is present at much lower percentages, from 1.5 to 4.3%. The authors attribute the lack of previous reports of these FA, in part, to the methods used [2]. However, these FAs were first identified in *Holothuria mexicana*, by a highly skilled team headed by Carballeira [46] as part of a continuing interest in lipid biochemistry of marine invertebrates. They identified a series of 2-hydroxy FAs, in particular 2-hydroxy-15-tetracosenoic acid (αOH 24:1n-9) that had been characterized previously by this research team as common in sea urchins [45]. 2-Hydroxylated MUFAs with a length of C22 and C24, in addition to MUFAs of identical chain-length, are the principal FAs of glycosphingolipids. The latter represent an unprecedented lipid class among the lipids isolated from sea urchins.

Accurate identification of the structure of these FAs is of high importance. As has been shown, at least, for sea urchins, the double-bond positions in the MUFAs 22:1, 23:1, and 24:1 are the same as those in the 2-hydroxylated αOH-22:1, αOH-23:1, and αOH-24:1. All of these acids belong to the n-9 family. These new hydroxylated FAs could have arisen from glucosylceramides, since they are not found in common phospholipids. As to their biogenesis, the possible pathways of their origin in holothurians are considered in Section 3. The origin and the role of αOH 24:1 and αOH 23:1 in holothurians remain to be determined.

New FAs, 7-methyl-6-octadecenoic acid (19:1) (0.7%) and 7-methyl-6-hexadecenoic (17:1) (0.4%), were identified in the phospholipids of *H. mexicana* [45]. The finding of both the E and Z isomers of 7-methyl-6-octadecenoic acid in a strain of *Vibrio alginolyticus* confirms the possible bacterial origin of this acid. In addition, the origin of 7-methyl-6-octadecenoic acid is most likely microbial, because the isomeric 10-methyl-9(Z)-octadecenoic acid was isolated from the marine fungus *Microsphaeropsis olivacea* [45].

The trophic preferences of holothurians or availability of food in their various habitats can have a significant effect on their FA profiles. As already emphasized above, different methodological approaches with different derivatization conditions and GC analysis of FAs, as well as difficulties in identifying FAs, can cause significant differences between the results of the FA analyses. To identify the dietary contribution to the composition of the holothurian FAs a PCA was performed (Figure 8). However, to minimize possible methodological discrepancies, we limited the consideration of FA data to only 16 holothurian species from three papers [38,75,76]. The results of the PCA indicate that the holothurian species clearly discriminated on the basis of their FAs composition. The PCA separates 15 species into five groups by their FAs, thus, suggesting difference in their diet (Figure 8). PC1 explains 56.37% of the total variance in the data. According to the PCA, 20:4n-6, 20:5n-3, 14:0, 20:0, and 23:1n-9 are the most important FAs for distinguishing between samples along PC1, whereas 20:5n-3, 20:4n-6, 20:1, 18:1n-7, 16:0, and OBFAs are most important along PC2. Figure 8 shows the projections of 16 variables significant for separation of holothurian species.

Group I comprises the species *Molpadia blakei*, *Cucumaria* sp., *Eupentacta fraudatrix* that possess the largest amounts of bacterial OBFAs (12–20%) along with elevated amounts of the SFAs 16:0 and 18:0 specific for detritus, as well as the highest content of 20:5n-3 (up to 37%) and the lowest percentage of 20:4n-6 (3–7%) among the analyzed holothurians species. Group II includes *Parariza prouhoi*, *Oneirophanta mutabilis*, *Psychropotes longicauda*, *Pseudostichopus villosus*, and *Deima validium* that are distinguished by an unusually high level of 23:1n-9 (11–18%) and an abundance of two major PUFAs, with almost equivalent proportions, 20:4n-6 (up to 28%) and 20:5n-3 (up to 26%), whereas OBFAs, SFAs, and MUFAs are minor components. *Amperima rosea* forms a separate group (III) due to an elevated level of 23:1n-9 (11%) and an abundance of 20:4n-6 (26.7%) with a moderate level of 20:5n-3 (14.5%). *Stichopus japonicus* alone forms a separate group (IV) that has a large amount of OBFAs (about 17%), 16:0, 18:0, and 16:1n-7 with moderate amounts of 20:4n-6 and 20:5n-3. Group V, consistsing of *Pearsonothuria graeffei*, *Bohadschia argus*, *Actinopyga lecanora*, *Holothuria moebii*, and *Holothuria pardalis*, shows a high percentage of 20:4n-6 (15-23%), 18:0 (5.2–9%), 16:0 (9–13%), and the highest level of 20:1 (7.1–16%) with OBFAs as minor components. Thus, the FA profiles of these species are clustered on the basis of feeding preferences of the animals and food availability in their habitats.

Most holothurian species are epibenthic sediment surface feeders that consume fresh benthic microalgae and detrital material from the upper few millimeters of the sediment surface [86]. It is, therefore, not surprising that they demonstrate a high proportion of PUFAs in their tissues. Undoubtedly, holothurians accumulate and store PUFAs when benthic microalgae and fresh phytodetritus are available, thereby providing themselves with these essential FAs vital for maintaining their vital functions, reproduction, and growth [86]. However, the FA profiles are more variable between species, thus, suggesting the differences in their food sources. The increased proportion of OBFAs, as well as 18:1n-7, specific for bacteria [67,68,69,70,90] in some species (from Groups I and IV) indicates that these holothurians are more reliant on bacterial food sources. The elevated level of 16:0 and 18:0 in the species from Groups IV and V suggest an increased proportion of detritus in their diet.

## 8. Fatty Acids of Echinoidea (Sea Urchins)

Representatives of the Echinoidea are widely distributed in the oceans from the tropical to polar zones and from rocky near-shore to abyssal depths. About 950 species of sea urchins are known to date; their main habitat is in sea beds with large benthic algae. Most biochemical studies have focused on the composition of the sea urchin gonads, because they are considered a highly valued seafood and delicacy in many parts of the world. The sea urchin gonads are unusual among marine invertebrates in their size; occupying large part of the body, they remain large before and during gametogenesis. Echinoid gonads play multitude of roles and are known to reflect both nutritive and reproductive status, which in turn, is manifested in their FA composition [91]. The vast majority of FA investigations have been performed on gonad tissues from various sea urchin species, with focus on seasonal changes and possible relationships with the reproductive cycle, in particular, from edible sea urchins that are of exceptional interest as seafood in human diet.

The class Echinoidea has a lipid composition dominated by EPA, followed by ARA, 16:0, and 20:1 (Figure 9, Appendix A). Unsaturated acids in them are more abundant than SFAs. Numerous PUFAs, such as 18:2n-6, 18:2n-3, 18:3n-6, 18:3n-3, 18:4n-3, 20:2n-6, 20:3n-3, 20:4n-3, 22:2n-6, 22:5n-3, 22:6n-3 constitute the largest class. EPA is reported to be a major PUFA in sea urchins. In almost all analyzed species, 20:5n-3 prevails over 20:4n-6, with their ratio ranging from 1.1 in *Strongylocentrotus droebachiensis* [92], *Paracentrotus lividus* [93], *Psammechinus miliari* [94] to 4.4 in *Strongylocentrotus franciscanus* [95], and even to 6.1 in *Scaphechinus mirabilis* [44]. Nevertheless, the prevalence of n−6 PUFA over n−3 PUFA is a general characteristic of the tropical species *Diadema setosum* from the coastal waters of Nha Trang Bay, central Vietnam [96], and some temperate species such as *Strongylocentrotus nudus* and *Pseudocentrotus depressus* [44]. In contrast, the level of DHA, which is considered one of the major FAs in marine invertebrates [8,62], is only a minor component varying from 0.3 to 2.5% of total FAs but reaching 5.6% in *Echinocardium cordatum* [44] and 4.8% in *Psammechinus miliaris* [91]. The SFAs 14:0, 15:0, 16:0, 17:0, 18:0, 20:0, and 21:0 make up the second largest class with a percentage ranging between 17.1 and 19.9%. Among MUFAs, 16:1n-7, 18:1n-9, 18:1n-7, 20:1n-9, 20:1n-7, 22:1n-9, and isomers of 20:1 are the especially important acids [66,92,94,97]. The origin of C20:1 may be biosynthetic since it is not reported for seaweeds, which is the natural diet of sea urchins [98,99]. Numerous NMI FAs are also found in sea urchin, with their content ranging from 2 to 13% between species [44,66,92,100]. Among them, the isomers of C20 NMI FAs, 20:2Δ5,11 and 20:2Δ5,13, are major. A series of 2-hydroxy FAs of the n-9 family, 22:1, 23:1, and 24:1, is reported for the Caribbean urchin, *Tripmaster esculentus*, and is known to occur in glycosphingolipids [46]. Some exotic FAs such as 21:4n-7, 22:4n-8, 22:5n-5, and 23:4n-9, earlier discovered in deep-sea foraminifera, and 22:6n-2 have been detected as minor components in the abyssal species *Kamptosoma abyssale* collected at depths of 5,210 and 6,183 m from the Kuril–Kamchatka Trench [41].

A predominant FA of *Psammechinus miliaris* for all collection localities and depths is 20:5n-3, ranging between 15 to 19%, followed by 20:4n-6 (11–14%) and 16:0 (about 10%) [102]. In lipids of soft tissues of *Tripneustes gratilla* and *Echinus esculentus*, 16:0, 20:4n-6 and 20:5n-3 are also the primary FAs [102]. The acids 16:0, 20:1, and 20:5n-3 are the main in *Strongylocentrotus droebachiensis* [66,92]. An overall pattern similar to that observed for *Psammechinus miliaris* was reported by Cook et al. [94]. Total PUFAs make up two thirds of total FAs in the gonads of *Strongylocentrotus intermedius*; among them, 20:4n-6 and 20:5n-3 are major, and together amount to about 30% of total FAs [103]. PUFAs make up high proportions in the gonads of *Paracentrotus lividus* from the Gulf of Tunis, constituting more than 40% of total FAs [100], and in *P. lividus* and *Arbacia lixula* from the southern coast of Spain [97] (Figure 9).

The biochemical composition of sea urchin gonads has been widely studied in different species. The studies have revealed its seasonal changes, possible relationship with the reproductive cycle, and influence of diet composition. The observed differences between the species arise mainly from different habitats of the sampled populations, which suggests different food availability, and also from different stages of the reproductive cycle. Furthermore, one of the explanations for the different results for the same species may be methodological problems associated mainly with the techniques used for obtaining FA derivatives and the conditions of analysis and FA identification.

The three main factors that can be responsible for the modification of tissue FAs are the water temperature, which probably depends on seasons, the reproductive cycle, as also the diet and food availability. These factors are interrelated, since it is well known that sea urchins have an annual reproductive cycle influenced by photoperiod, temperature, and quality and quantity of available food [104]. Thus, the effect of environmental conditions including temperature variations on the FA composition of sea urchins should be discussed with regards to the reproductive cycles and nutrition, mainly the type of food that can also contribute to variation in FA composition in sea urchin gonads [100].

As has been reported for in marine invertebrates, as temperature decreases, there is an increase in PUFAs aimed to maintain the functionally optimal membrane fluidity. Though no seasonal variations in 16:0 and 22:6n-3 are observed in the gonads of *A. lixula* and *P. lividus*, 20:5n-3 markedly decreases as temperature rises, it suggests a possible involvement of 20:5n-3 in the adaptation of membrane lipids to high temperatures [97]. The highest PUFA levels in *P. lividus* were recorded in the winter and spring, while the lowest ones occurred in the summer. The seasonal variations in PUFAs in *P. lividus* are primarily due to the variations in 20:5n-3 and 20:4n-6 [100]. The gonad FAs in *P. lividus* and *A. lixula* vary significantly during the year; the patterns of these variations are similar for both species despite the marked differences in their feeding habits [97]. The authors conclude that two main factors could be responsible for these seasonal variations: fluctuations in seawater temperature and the annual gametogenic cycle.

The lipid and FA profiles of gonads change with season [97,101,105,106,107] or gametogenic stage [91,108]. The lipid and FA profiles also differ between sexes, with ovaries showing greater, than testes, accumulation of lipids [101,105] and storage lipids, in particular, triacylglycerol providing the energy source for early larval development. Differences in FA composition between male and female gonads of *Paracentrotus lividus* and *Arbacia lixula* may reflect the metabolic peculiarities of gonad tissues related to specific requirements for spermatogenesis and oogenesis [97]. The differences in the FA composition of the gonads reflect also the changes in gonad maturation [91,97,101].

Sex is the major factor affecting the FA profile of the gonads (intact and spawned) and gametes in *Arbacia dufresnii*, sex-related differences are observed in the variety of FA, their total concentration, and proportions [109]. There is a significant difference in the FA profile between intact ovaries, spawned ovaries, and eggs. Eggs have a higher proportion of unsaturated FAs than sperm, which is dominated by SFAs. However, both eggs and sperm of *A. dufresnii* contain a high percentage of two essential FAs, 20:4n-6 and 20:5n-3. The differences in the FA profile between mature ovaries and testes of sea urchins presumably result from the differences in the FA composition of oocytes and sperm, which are at maximum levels prior to spawning [91,97,101].

The relationship between the gonad FAs and the reproductive cycle has been demonstrated for various sea urchin species. Major changes in the FA composition of the gonads occur during sexual maturation, with 20:5n-3 and 22:6n-3 being particularly important in this maturation cycle. Mature males and females differ significantly in the FA composition of their gonads, whereas post-spawned individuals show no gender differences. Males have higher level of PUFAs compared to females and show a dramatic reduction in 22:6n-3 and 20:5n-3 with the succession of maturity stages [91]. Differences in the FA profile of oocytes and sperms were observed in numerous species, including *Arbacia dufresnii* [109] and *Strongylocentritus intermedius* [110], *Psammechinus miliaris* [91], *P. lividus*, *A. lixula* [97], and *Arbacia dufresnii* [101].

Since the gonads are both the reproductive organ and a nutritive store, any observed differences in the FAs may be attributed not only to the changes in maturity over time, but also to different diets of the animals [91,111,112]. Animals are known to receive a considerable amount of FAs from their food [64,66] and, therefore, diet has a major effect on their FA composition. Echinoidea species are generalist grazers whose feeding ecology is well known from field studies and analyses based on the FA biomarker approach [66,102,111,113]. The diet of sea urchins varies between habitats and to depend on food availability. They are able to exploit a number of food sources, although brown macroalgae constitute the main trophic resource for them [66,101]. Sea urchins feed primarily on macroalgae but also consume benthic diatoms [67], microorganisms, and bacteria from mud [94,95], and even slow-moving or sessile animals [113]. The relationship between the FA composition of gonads and diets of sea urchins has been documented in many studies [36,66,94,105,111,112,113,114,115,116].

Brown algae have higher proportions of C18 PUFAs (18:4n-3, 18:2n-6, 18:3n-3), 20:4n-6, 20:5n-3, 18:1n-9, and 16:0 [98,99]. Therefore, the higher levels of these FAs found in the sea urchin gonads actually reflect the FA content of brown algae, and the significant content of C18 PUFA (Figure 9, Appendix A) is considered an indicator of brow algal diet. The effect of environmental conditions on the gonad FA composition of *P. lividus* is probably related to the differences in available food resources [97,111]. This species usually feeds on brown and, rarely, green algae. Some of the sampling sites are dominated by the brown alga *Stypocaulon scoparia* and the other by an assemblage of the brown algae *Dyctiota* spp. and the green algae *Ulva* spp. Changes in the gonad FAs of *P. lividus* are more probably related to the differences in the macroalgae species that the sea urchins have consumed at both sites [97]. The FA composition of sea urchin *P. miliaris* feeding on mud differ from that of the sea urchin feeding on algae, and the difference is attributed to the sediment components of their diet [94]. The high contribution of OBFAs in FAs of the mud-feeding sea urchin, *Strongylocentrotus franciscanus* [95], reflects the relatively high contents of these bacterial FAs in the bottom mud. The FA profiles of the irregular sea urchins *Spatangus purpureus* agree well with the FA compositions of the potential trophic resources (red algae and sediment) and indicate changes between localities with different available food resources. Namely, EPA and ARA being abundant in red macroalgae mainly contribute in FAs of the gonads. Trophic markers of bacterial input (high proportion 18:1n-7) and carnivorous feeding (large proportion of DHA) are significantly more abundant in sea urchins collected from bottoms with poor vegetation. [113].

Nevertheless, the ability of sea urchins to biosynthesize PUFAs attracts a lot of attention. It is assumed that sea urchins can modify the dietary precursors, such as linoleic acid 18:2n-6 and α-linolenic18:3n-3 and biosynthesize 20:4n-6 and 20:5n-3, respectively [94,111,116]. This has become evident from feeding experiments with *P. lividus* larvae fed various microalgae species. The larvae had higher levels of 20:4n-6 and 20:5n-3, although these FAs were at relatively low levels in their diets and absent in the microalgae. DHA 22:6n-3 was also present in the larvae, although it was not found in *Dunaliella tertiolecta* [112]. The higher concentrations of PUFAs observed in sea urchin tissues compared to those in algae or other experimental diet suggest that sea urchins are able not only to accumulate FAs, but also to synthesize PUFAs [94,117]. For example, *P. miliaris* has the ability to synthesize small amounts of 20:5n-3 from 18:3n-3 among other PUFA modifications [117]. Castell with coauthors found that *S. droebachiensis* can synthesize 20:4n-6 from 18:2n-6 supplied with the diet. The level of 20:5n-3 has a direct relationship with the 18:3n-3 precursor in the diet [111]. *Strongylocentrotus droebachiensis* and *S. purpuratus* also synthesize and accumulate ARA and EPA in the gonad and gut tissues [116]. Bioconversion of 18:3n-3 to EPA and 18:2n-6 to ARA was also assumed for *P. lividus* [118]. In addition, a high proportion of 20:2Δ5,11 was recorded in the gonad and gut tissue of *S. droebachiensis* (6–9% in the gonad and gut) [119] and *S. purpuratus* (up to 8–9 in the gonad and gut) [116].

The ability of sea urchins to biosynthesize LC-PUFA can be confirmed by the molecular and functional characteristics of fatty acyl desaturases (Fads), key enzymes in the biosynthesis of LC-PUFA, in the sea urchin *P. lividus* [120]. The functional characteristics of Fads have confirmed the presence of a Δ5 desaturase with activity towards SFAs and PUFAs. This Fad (FadA) plays a role in the NMI FA biosynthesis. Another Fad (FadC) showed Δ8 activity. The authors conclude that *P. lividus* possesses desaturases that are responsible for all desaturation reactions required to biosynthesizes the essential EPA and ARA through the so-called “Δ8 pathway” [120].

## 9. Echinoderm Fatty Acids as Bioactive Compounds

FAs are structural components of membrane phospholipids that influence the cell functions via effects on the membrane properties [121]. FAs are associated with a wide range of bioactivities including antioxidant [122], antimicrobial [123], antifungal and antimalarial [124], anti-diabetic [125], and anticancer activities [126]. They act as a precursor pool for lipid mediators [127]. FAs are also essential as substrates for the formation of eicosanoids, a group of biologically active molecules known as local hormones, which include prostaglandins, thromboxanes, and leukotrienes, and also more recently identified lipid mediators [127]. Eicosanoids are formed through oxygenation of n-3 and n-6 C20 PUFAs. An elevated level of dietary n-6 relative to n-3 has been found to correlate with human cardiovascular diseases, hypertension, and autoimmune disorders.

PUFAs, mainly of the n-3 family, are of particular interest as well-known essential lipids necessary for maintaining human health and balanced nutrition. Echinoderms represent a major source of PUFAs. The n-3 PUFAs influence a variety of health disorders such as hypertension, heart disease, diabetes, cancer, and allergic diseases by modulating membrane constitution, inhibiting production of proinflammatory eicosanoids and cytokines, and binding to cell surface and nuclear receptors.

The n-3 PUFAs constitute a heterogeneous family, of which the most biological active acids are EPA and DHA mainly found in marine fish and invertebrates as an edible source [128]. They are involved in numerous physiological regulation mechanisms and have antioxidant and anti-inflammatory properties. These PUFAs are essential for the development of the nervous system, brain, and retina. Their effects on health are globally recognized as beneficial [129]. The n-3 PUFAs may relieve some psychiatric and neurodegenerative diseases, as evidenced by both experimental and clinical studies [130]. Dietary n-3 PUFAs decrease the risk of metabolic diseases including diabetes mellitus [125].

PUFAs are increasingly attracting attention as new potential topical agents for the treatments of infections caused by gram-positive bacteria due to their antimicrobial potency and anti-inflammatory properties. In study of antimicrobial activity of two n-3 FAs, DHA and EPA using an in vitro multi-species subgingival biofilm model, both DHA and EPA showed a significant activity against six bacterial species (*Streptococcus oralis*, *Actinomyces naeslundii*, *Veillonella parvula*, *Fusobacterium nucleatum*, *Porphyromonas gingivalis*, and *Aggregatibacter actinomycetemcomitans*) included in this model [131].

Of particular interest is the nisinic acid, 24:6n-3 (THA), which is specific for echinoderms such as, in particular, sea lilies and ophiurans. It is a very-long chain FA with a chain length greater than C22. Belonging to the n-3 family, it has been attributed to essential FAs [132]. THA is present in mammalian tissues, including spermatozoa, the retina, and the brain, along with other very long chain FAs [133,134]. THA is believed to be important as an intermediate in 22:6n-3 biosynthesis [37]. Dietary and stable isotope studies on rats have confirmed that the C24 n-3 PUFAs, 24:5n-3 (TPA), and 24:6n-3, are immediate precursors to 22:6n-3 in the biosynthetic pathway [134]. Even though 24:6n-3 is a PUFA existing in mammalians, its physiological functions have been poorly studied. THA, as well as other PUFAs, has shown beneficial properties for human health. A study of the human n-3 PUFA metabolism has revealed that plasma TPA and THA levels are higher in women and increase with supplementation of EPA, but not DHA, suggesting an accumulation of THA prior to the conversion to DHA in the n-3 PUFA synthesis pathway [135].

Ishihara with coauthors [136] purified 24:6n-3 from the brittle star *Ophiura sarsi* and found that it could inhibit the antigen-stimulated production of leukotriene-related compounds such as LTB4, LTC4, and 5-hydroxyeicosatetraenoic acid, as well as other n-3 PUFAs, 20:5n-3 and 22:6n-3, in a mouse mast cell line. It also could reduce the histamine content in MC/9 cells [136]. The patterns of the effects of 24:6n-3 on the eicosanoid synthesis and histamine content are more similar to those of 22:6n-3 than 20:5n-3. As the results showed, anti-inflammatory and antiallergic activities of 24:6n-3 similar to those of 22:6n-3 can be expected. The anti-inflammatory and antiallergic effects of 24:6n-3 are noteworthy because these are closely related to the PUFA metabolism.

A comparison of the effect of n-3 PUFAs (20:5n-3, 22:5n-3, 22:6n-3, and 24:6n-3) on lipid metabolism of HepG2 cells has shown that THA has the strongest ability to suppress the synthesis of triacylglycerols and cholesteryl ester among these n-3 PUFAs [137]. The highest activity of THA in suppressing the hepatic triacylglycerols accumulation and increase in liver weight has been confirmed on C57BL/KsJ-db/db mice [138]. The authors conclude that clinical functions of n-3 high unsaturated FA is related to the number of double bonds and carbon atoms in their structure. According to these results, THA is a valuable component capable of reducing the hepatic lipid accumulation and having a high potential as a functional compound beneficial for human health [77].

N-acylethanolamines (NAEs) are endogenous lipid-signaling molecules derived from FAs that regulate numerous biological functions, including those in the brain. Tetracosahexaenoylethanolamide (THEA), a NAE product, has been shown to be a new NAE produced in the mouse brain at elevated levels upon ischemia/hypercapnia and regulating neuronal excitation, which suggests a neuroprotective response [139].

One of therapeutic approaches for preventing diabetes mellitus and obesity is to prevent the absorption of glucose via inhibition of α-glucosidase. Two unsaturated FAs purified and identified from the body wall of *Stichopus japonicus*, 7(Z)-octadecenoic acid (Δ7-18:1) and 7(Z),10(Z)-octadecadienoic acid (Δ7,10-18:2), exhibit strong α-glucosidase inhibitory activity. Therefore, FAs derived from sea cucumbers can potentially be used to develop novel natural nutraceuticals for the treatment of type-2 diabetes [140].

Echinoderms contain several unusual and exotic FAs having various bioactivities. Among them, a rare PUFA, 21:4n-7, has been recorded from six species of deep-sea holothurians collected in the Sea of Okhotsk waters off the Kuril Islands [42] and from a deep-sea sea star and a sea urchin from the Kuril–Kamchatka Trench [41]. This acid shows high cytotoxic activity against a range of human cancer cell lines (melanoma, colon carcinoma, and breast carcinoma). The n-7 PUFA is significantly more cytotoxic compared to the n-6 ARA and n-3 EPA. A significant non-selective cytotoxicity of 21:4n-7 against normal lung fibroblasts has also been reported [141].

Another group of uncommon FAs discovered in echinoderms is hydroxy FAs. OH-FAs are very important chemicals used for a variety of applications such as production of biodegradable materials and also cosmetic and pharmaceutical polymers. These are difficult to be synthesized via chemical routes due to the inertness of the fatty acyl chain [142]. In addition to being the major class of natural products in bacteria, yeasts, and fungi, these acids have been found in some of echinoderm species. In sea urchins, αOH 22:0, αOH 22:1, αOH 23:0, αOH 23:1, αOH 24:0, and αOH 24:1 were identified [46]; αOH 18:0 in a holothurian species [45]; αOH 23:1 and αOH 24:1 were detected in holothurians from the abyssal zone [2]. Monohydroxy octadecenoic acid, OH-18:0, exhibits antifungal activity [143]. The mechanisms responsible for the antifungal effect of OH-FAs still remain unknown. One general mechanism suggested for antifungal FAs is that based on the detergent-like properties of the compounds, affecting the structure of cell membranes of the target organisms. This increases the membrane permeability and the release of intracellular electrolytes and proteins and, eventually, leads to the cytoplasmic disintegration of fungal cells.

NMI FAs are found in a wide variety of marine animals, and, therefore, can be attributed to common FAs. However, these are different from usual FAs and belong to the FAs with a series of double bonds in which at least one adjacent pair of double bonds is separated by at least two carbon atoms, i.e., by a group other than a single methylene group. These FAs are variants on methylene-interrupted structures: 5,11,14,17-eicosatetraenoic acid and 5,11,14-eicosatrienoic acid. Members of the classes Asteroidea and Echinoidea are especially rich in NMI FAs of different structures. In addition to 20:2Δ5,11 and 20:2Δ5,13 that are typical of marine invertebrate lipids and their homologues 22:2Δ7,13 and 22:2Δ7,15, echinoderms also have unusual cis-5-olefinic compounds, e.g., 20:3Δ5,11,14, and 20:4Δ5, 11,14,17, which were first identified in the polar lipids of 12 sea urchin species from Japan and Canada [44]. Other echinoderm classes can be arranged in the order of decrease in the cis-5-olefinic acids content is as follows: sea urchins, sea stars, ophiuroids, sea cucumbers, and sea lilies. The specific double bond positions in NMI FAs are believed to confer to cell membranes a higher resistance to oxidative processes and microbial lipases than common PUFAs [63].

NMI FAs, in particular, sciadonic acid, cis-5,11,14-eicosatrienoic acid or 20:3Δ5,11,14, are recently discovered functional lipids with potential pharmacological effects. Sciadonic acid is an anti-inflammatory FA displacing ARA from specific phospholipid pools, thus, modulating downstream pro-inflammatory lipid mediators. The antioxidant and anti-inflammatory properties of NMI FAs were proven in experiments. The anti-inflammatory and antioxidant effects of sciadonic acid, inhibiting the activity of phosphodiesterase and lipoxygenase-5, have been clearly demonstrated [144]. An in vivo study confirms that sciadonic acid markedly inhibits expression and activity of stearoyl-CoA desaturase 1, the hepatic Δ9-desaturase involved in the formation of MUFAs, which demonstrate promising effects on hyper-triglyceridemic models [145]. It has been shown that sciadonic acid can modify lipid metabolism in rats: when rats were fed sciadonic acid, the serum and liver triacylglycerol levels decreased [146].

The introduction of a trans-5 double bond into the molecule of linoleic acid, 18:2n-6, results in columbinic acid, 18:3Δ5trans,9cis,12cis, having attractive biological properties [147]. Experiments with essential FA-deficient rats, have shown that the columbinic acid is effective in maintaining the proper epidermal layer and improves the fertility of rats, while the inhibition of prostaglandin synthesis has a beneficial effect, since inflammation and the thrombotic tendency are reduced [148].

Recently, it has been reported that the sciadonic acid under controlled clinical conditions has an anti-inflammatory potential in human skin accelerates irritant-induced healing, and improves the skin barrier function. Improvement in the barrier function would benefit the treatment of dermatitis, acne, eczema, and skin scarring. Oil with Δ5 NMI FAs has a potential to maintain healthy, moisturized skin and to improve the skin structure, elasticity, and firmness [148].

Autoimmune diseases can be controlled by the administration of NMI FAs. A method for suppressing autoimmune diseases in a mammalian has been developed that consists in administering a therapeutically effective amount of NMI compounds [149]. One of the approaches to treating autoimmune diseases was based on studies of the role of cell membranes. PUFAs are known to be essential for the immune function but only at very high concentrations and, therefore, the focus on n-3 PUFAs have not yielded success. It has been found that the formation of autoantibodies and, hence, the onset of autoimmune diseases can be suppressed by the use of NMI FAs. These acids are variants on methylene-interrupted structures: 20:3Δ5,11,14 and 20:4Δ5, 11,14,17. A preferred compound is the sciadonic acid, Δ5,11,14-eicosatrienoic acid. Recent studies have demonstrated that trienoic acids with NMI cis-double bonds show moderate cytotoxic activities against tumor cell lines (Jurkat, K562, U937, HL60, HeLa), human embryonic kidney cells (Hek293), normal fibroblasts, and human topoisomerase I (hTop1) inhibitory activity in vitro [150].

The ethyl acetate extracts from the sea cucumber *Holothuria leucospilota* showed the highest antibacterial activity against *Staphylococcus aureus* and antifouling activities. The major compounds of the ethyl acetate extract of *H. leucospilota* were FAs and terpenes that may be responsible for the antibacterial and antifouling activity [151].

Crinoids are the least studied class of echinoderms as regards their bioactivity. The pharmacological potential of the feather stars *Comaster schlegelii* and *Himerometra robustipinna* has been evaluated recently. The extracts of these animals have shown a significant antibacterial activity against human pathogens, and also anti-algal and antioxidant activities. The anti-diabetic potential of the extracts has been screened by evaluating their ability to modulate the enzyme α-amylase. According to the results, the extracts of both species have a moderate inhibitory effect on this enzyme. A GC–MS analysis has detected the presence of FAs and sterols as the major bioactive compounds that may be attributed to the bioactivities. The study shows crinoids as a promising source of bioactive compounds with a high pharmacological potential [12].

Thus, many FAs, both common and unusual, found in the Echinodermata have a range of biological activities and are a source of functional lipids with potential pharmacological effects.

## 10. Conclusions

The Echinodermata is a phylum of marine invertebrate constituting an important component of the invertebrate fauna of the oceans. They are distributed in almost all latitudes, environments, and depths, from the intertidal to abyssal zones depths, and play a variety of ecological roles. Echinoderms are a unique source of bioactive compounds from different classes of natural substances. As many as approximately a third of natural products of marine origin have been derived from echinoderms. The lipid chemistry of echinoderms has been the subject of numerous attempts to discover new metabolites with intriguing biological activities. FAs are reported to be associated with a wide range of bioactivities including antioxidant, antimicrobial, antidiabetic, and anticancer ones. They act as a precursor pool for lipid mediators. FAs are structural components of membrane phospholipids that influence the cell functions via effects on the membrane properties.

To date, extensive information has been obtained on the FAs of the representatives of five echinoderm classes: Crinoidea (sea lilies and feather stars), Holothuroidea (sea cucumbers), Echinoidea (sea urchins), Asteroidea (sea stars or starfish) and Ophiuroidea (brittle stars). Members of this phylum exhibit a variety of FA structures, which include common, unusual, and even exotic FAs. The own biosynthetic capabilities of echinoderms and specificity of their diet lead to the emergence and accumulation of a number of unusual FAs. Certain common and unusual FAs or FA groups may serve as chemotaxonomic markers of the classes within the phylum. The specificity of FA biosynthesis and composition in different taxonomic groups provides fertile ground for their wide application as chemotaxonomic and biochemical markers of trophic and metabolic interactions in aquatic ecosystems. Diet, growth, reproductive stages, and environmental conditions exert strong influences on FA profiles. According to the lifestyle of echinoderms, their feeding modes can be suspension feeding, deposit feeding, and predation; some species use more than one mode to obtain food. Thus, the FA profiles of echinoderms can be significantly influenced by their dietary preferences or by the availability of food in various habitats.

In general, echinoderms, like other marine organisms, are rich in essential PUFAs, mainly EPA, AA, and, to a markedly lesser extent, DHA. The content of 20:5n-3 prevails over that of 20:4n-6 in Ophiuroidea, while the opposite pattern is observed in Holothuroidea and Echinoidea. Besides common PUFAs, a very long-chain FA, tetracosahexaenoic or nisinic acid 24:6n-3, is characteristic mainly of Crinoidea and Ophiuroidea. The uncommon acid 23:1n-9 occurs almost only in Holothuroidea, and, therefore, this acid can be considered an important chemotaxonomic marker of this class. The unusual isomer 20:1n-13 is prominent in lipids of Ophiuroidea and Asteroidea and is not found in Crinoidea. A characteristic feature of the FA composition of echinoderms is the presence of rare or exotic PUFAs, especially in deep-sea species. The most noteworthy FAs are a series of 2-hydroxylated FAs found in holothurians and sea urchins. Members of the Asteroidea and Echinoidea are especially rich in NMI FAs with different structures. Among them, cis-5,11,14-eicosatrienoic or sciadonic acid 20:3Δ5,11,14 is of particular interest as it exhibits a wide range of biological activities. Both common and unusual FAs detected in the members of the phylum Echinodermata show various biological activities, and, therefore, are considered a source of functional lipids with potential pharmacological effects.

## Figures and Tables

**Figure 1 marinedrugs-21-00021-f001:**
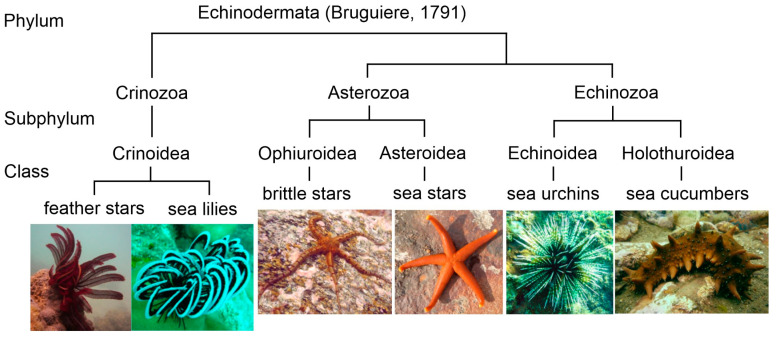
Cladogram of the phylum Echinodermata. The photos are reproduced with permission from Konstantin Dudka.

**Figure 2 marinedrugs-21-00021-f002:**
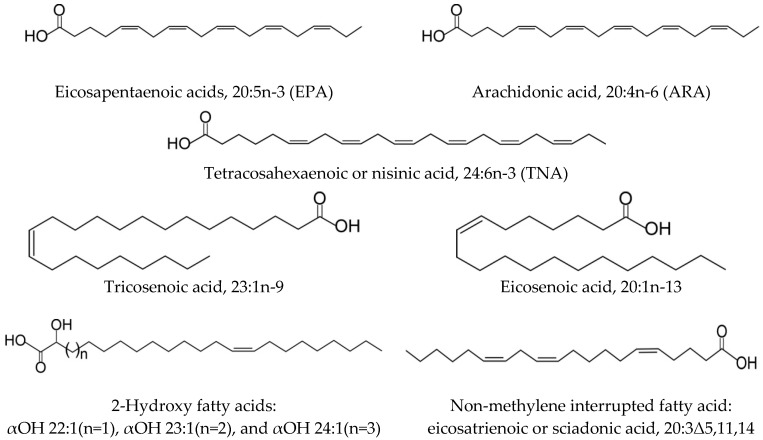
Structures of fatty acids typical for echinoderms.

**Figure 3 marinedrugs-21-00021-f003:**
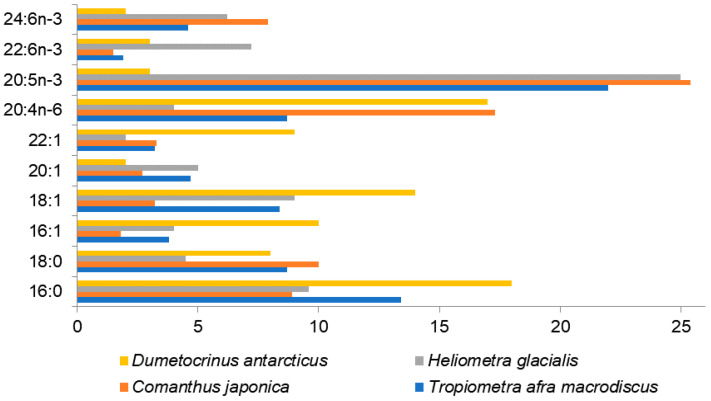
The fatty acid composition of Crinoidea (wt%). Data compiled from [47,73,74].

**Figure 4 marinedrugs-21-00021-f004:**
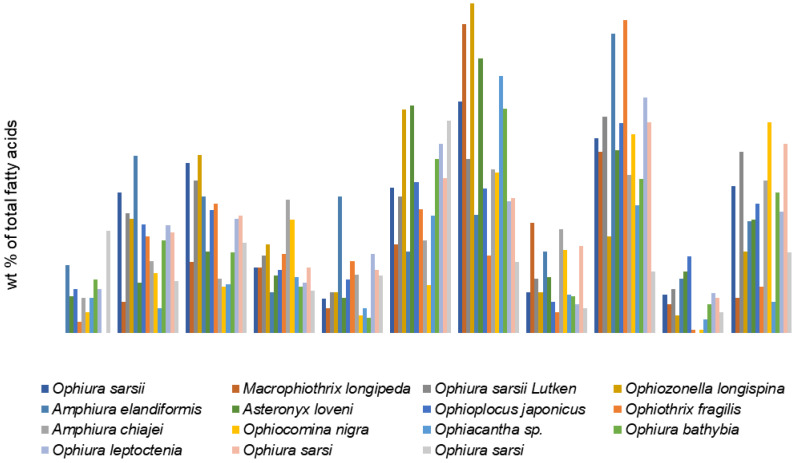
The fatty acid composition of Ophiuroidea (wt%). Data compiled from [2,37,41,47,53,73,77,78].

**Figure 5 marinedrugs-21-00021-f005:**
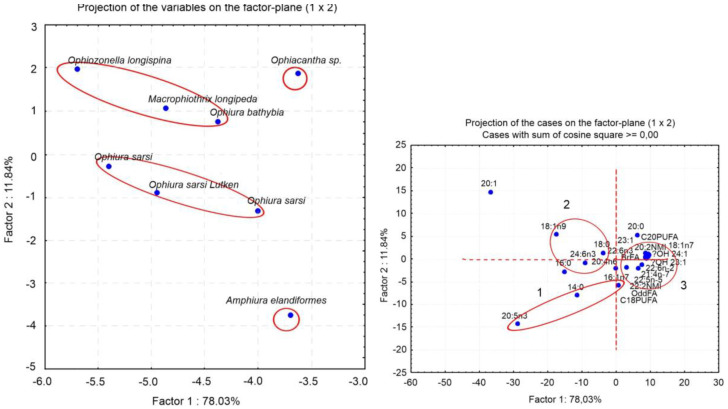
Principal component analysis (PCA) of fatty acids of eight Ophiuroidea species. Data compiled from [2,41,77].

**Figure 6 marinedrugs-21-00021-f006:**
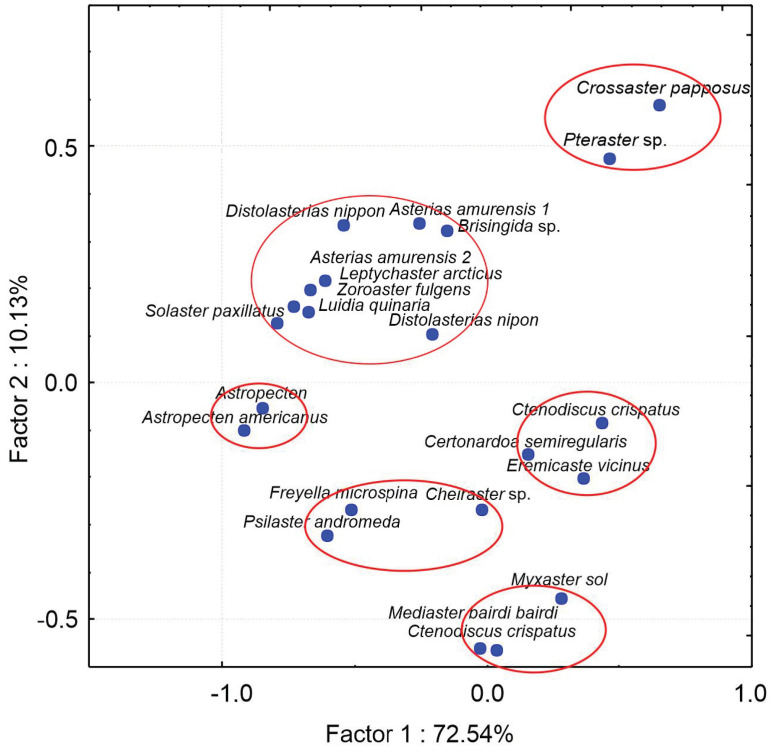
Principal component analysis (PCA) of fatty acids of 21 Asteroidea species. Data compiled from [9,77,79,85].

**Figure 7 marinedrugs-21-00021-f007:**
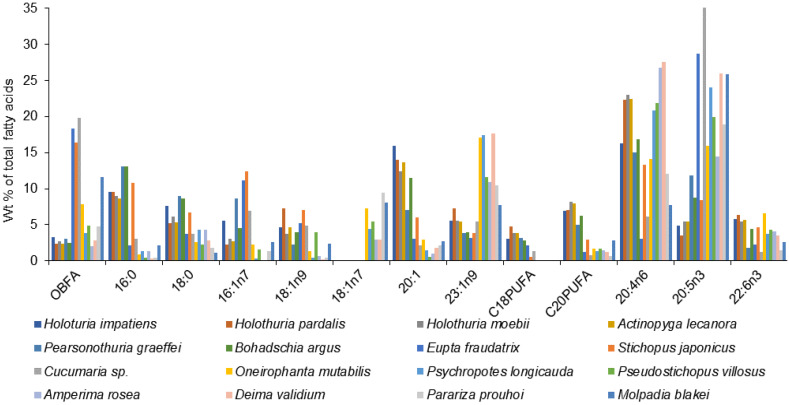
The fatty acid composition of Holothuroidea (wt%). Data compiled from [59,75,76].

**Figure 8 marinedrugs-21-00021-f008:**
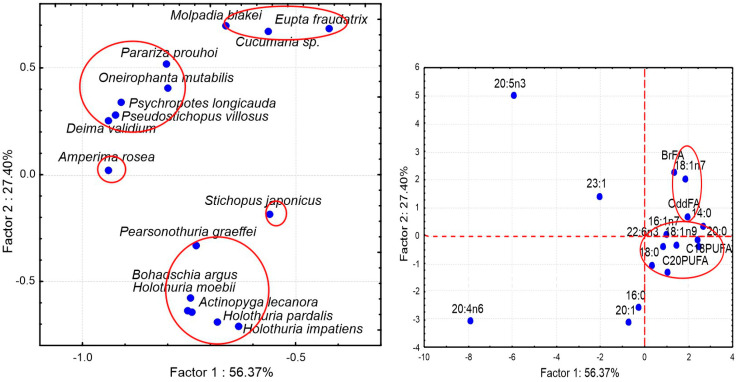
Principal component analysis (PCA) of fatty acids of 16 Holothuroidea species. Data compiled from [2,38,75,76,86].

**Figure 9 marinedrugs-21-00021-f009:**
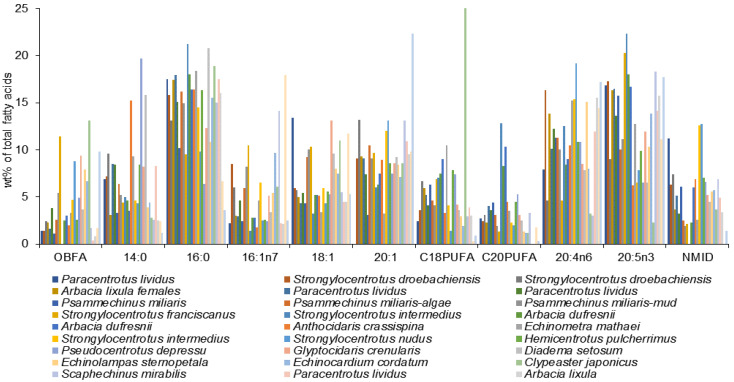
Distribution of the major fatty acids in Echinoidea. Data are compiled from [9,36,43,44,66,91,93,95,97,101].

**Table 1 marinedrugs-21-00021-t001:** Specific fatty acids identified in Echinodermata.

Fatty Acids	Species	Methods	Reference
24:6n-3	brittle star *Amphiura elandiformis*	GC-MS of FAMEs, DMOX derivatives	[37]
20:1n-13	brittle star *Amphiura elandiformis*	GC-MS of DMDS adducts; silver-ion chromatography with a solid-phase extraction	[37]
23:1n-9	sea cucumber *Stichopus japonicus, Cucumaria* sp., *Holothuria leucospilota*	GC of MEFAs, OTMS derivatives, pyrrolidides	[38]
20:2Δ5,11; 20:2Δ5,13; 22:2Δ7,13; 22:2Δ7,15	starfish *Asterias vulgaris*	AgNO_3_-TLC, GC of MEFAs	[39]
Δ7E,13E-20:2 Δ7E,13E,17Z-20:3 Δ9E,15E,19Z-22:3 Δ4Z,9E,15E,19Z-22:4	brittle star *Ophiura sarsi*	GC-MS of DMDS adducts, hydrogenation and AgNO_3_-TLC GC of the monoenoates.	[40]
21:4n-7, 22:4n-8, 22:5n-5, 23:4n-9	sea star *Eremicaster vicinus*	GC-MS of DMOX derivatives	[41]
21:4n-7, 22:4n-8, 22:5n-5, 23:4n-9	sea urchin *Kamptosoma abyssale*	GC-MS of DMOX derivatives	[41]
Δ5,8,11,14,17,20-22:6 or 22:6n-2	sea star *Eremicaster vicinus* sea urchin *Kamptosoma abyssale*	GC-MS of DMOX derivatives	[41]
21:4n-7	holothurians *Chiridota* sp., *Molpadia orientalis*, *Pseudostichopus mollis*, *Synallactes nozawai*	GC-MS of DMOX derivatives	[42]
26:7n-3, 26:6n-3, 26:5n-3	5 species of Ophiuroidea: *Ophiopholis aculeata* *Ophiopenia vicina* *Amphiophiura ponderosa* *Ophiura leptoctenia* *Asteronyx loveni*	Hydrogenation, GC-MS of MEFAs and DMOX derivatives	[43]
18:1Δ5; 20:1Δ5; 20:2Δ5,11;20:2Δ5,13; 20:3Δ5,11,14; 20:4Δ5,11,14,17	12 species of Echinoidea	AgNO3-TLC, reductive ozonolysis, GC-MS of the ozonolysis products. ^13^C-NMR.	[44]
7Me-Δ6Z-19:1 7Me-Δ6E19:1	holothuria *Holothuria mexicana*	GC-MS of pyrrolidides and total synthesis	[45]
2OH-Δ13-22:1 2OH-Δ14-23:1 2OH-Δ15-24:1	sea urchin *Tripneustes esculentus*	GC-MS of DMDS adducts, hydrogenation, ‘H NMR	[46]
2OH-Δ15-24:1	holothuria *Holothuria mexicana*	GC-MS of pyrrolidides and total synthesis	[45]

Abbreviation: DMDS: dimethyl disulfide, DMOX: 4,4-dimethyloxazoline; OTMS: trimethylchlorosilane; GC-MS: gas chromatography–mass spectrometry; MEFAs: methyl esters of fatty acid; TLC: thin layer chromatography.

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
