# Peer review of "Fatty Acids of Echinoderms: Diversity, Current Applications and Future Opportunities"

_marinedrugs, 2022, doi:10.3390/md21010021_

Round 1

Reviewer 1 Report

In this article " Fatty Acids of Echinoderms: Diversity, Current Applications and Future Opportunities", author mentioned about the Diversity, Current Applications and Future Opportunities regarding to Fatty Acids of Echinoderms. Overall, based the data provided, it can be considered for publication in this journal. However, there are some issues that have to be fixed before publication;

For increasing the interest of readers, the abstract could be more specific in term of scope, objectives and conclusions.

Please specify the novelty, significance, technical merit of the study in a better way.

English is very poorly presented throughout the manuscript. as well as the presentation is also very poor, should be uniform.

Figures should be uniform like an irregularity was found in Figure 1.

The paragraph distribution should be based the content of text like 176-179, only few lines.

Please revised introduction with proper consequences.

I am not sure for the permission as the author mentioned data compiled from. please check.

Besides putting the information which were previously published, authors need to write some critical statements on these properly.

Please add one section of application separately for better understanding.

Some errors regarding the sub/super script, spacing and typo need to consider throughout the manuscript.

Make sure that the format of references are uniform. Moreover, please add some more references to support this study.

Author Response

I would like to express my sincere gratitude to the Reviewer for the attention to my work and for the comments provided.

In this article " Fatty Acids of Echinoderms: Diversity, Current Applications and Future Opportunities", author mentioned about the Diversity, Current Applications and Future Opportunities regarding to Fatty Acids of Echinoderms. Overall, based the data provided, it can be considered for publication in this journal. However, there are some issues that have to be fixed before publication;

Q: For increasing the interest of readers, the abstract could be more specific in term of scope, objectives and conclusions.

A: In my opinion, in the abstract, I confirm the scope, objectives and conclusions are clearly defined. I highlighted all important positions of the review. The Reviewer's recommendations are too general for me to follow them. If additional information and details are required, please let me know what exactly is missing.

Q: Please specify the novelty, significance, technical merit of the study in a better way.

A: These aspects are defined at the end of Introduction: “Numerous reviews have been published on various bioactive natural compounds derived from echinoderms. In regards to echinoderm lipids and fatty acids (FAs), no reviews on this topic are currently available.”

The present review is based on consideration of the most interesting advances in the study of echinoderm FAs. Thus, without covering the subject exhaustively, this review makes emphasis on the most important factors and effects that determine the biodiversity of the echinoderm FAs, with focus on the crucial relationship of FAs with the biosynthetic capacities of animals and with their food spectrum. The major goals of this review were to illustrate the molecular biodiversity of FAs in echinoderms and distribution of FAs over the classes of this phylum, as well as to identify the most important FAs with bioactive potential.”

Q: English is very poorly presented throughout the manuscript. as well as the presentation is also very poor, should be uniform.

A: English was revised by a native speaker.

Q: Figures should be uniform like an irregularity was found in Figure 1.

A: There are 9 figures in the review. There are following types if figures: a cladogram of classes with photos of animals, chemical formulas, histograms with the distribution of the main fatty acids among classes of Echinoderms and figure with the results of the principal component analysis. They are uniform within their type but, each of the figures has its distinct purpose and thus can’t be uniform. In addition, Technical Editors are verifying this further down the track.

Q: The paragraph distribution should be based the content of text like 176-179, only few lines.

A: It has been corrected.

Q: Please revised introduction with proper consequences.

A: Considering the importance and relevance of the topic should be justified, in the introduction I described the biodiversity of echinoderms, their widespread occurrence in the marine environment, and their importance to humans in terms of food, medicine, and scientific research. Echinoderms are a rich source of bioactive substances of various chemicals. In this regard, numerous reviews have been published, while there are no reviews for echinoderm fatty acids. Meanwhile, a large amount of information has been accumulated on this taxonomic group, which, along with the well-known n-3 PUFAs, has a number of unusual components with bioactivity. Further aims were formulated.

Q: I am not sure for the permission as the author mentioned data compiled from. please check.

A: In the review, I’ve only used the published data, and these papers are referred. There’s no copyright violation.

Q: Besides putting the information which were previously published, authors need to write some critical statements on these properly.

A: I share the Reviewer's point of view where in addition to summarizing the information, the review should contain the author's own understanding and analysis of the data. That is why the principal component analysis of the fatty acids has been done for classes of echinoderms. It proves fatty acid composition depends significantly on the diet of the animals. To prove the point even further, I supplemented each section with an extensive discussion on the relationship of the fatty acid composition of animals with their diet. Discussion of pathways of biosynthesis of fatty acids was based on both previously published data and the modern information about the molecular and functional characteristics of fatty acyl desaturases. In addition, I emphasized the variability of the results of different authors is often and largely determined by differences in the conditions of GC analysis and inaccuracy in the identification of fatty acids.

Q: Please add one section of application separately for better understanding.

A: The section 9 is devoted to the application.

Q: Some errors regarding the sub/super script, spacing and typo need to consider throughout the manuscript.

A: Errors have been corrected in the text.

Q: Make sure that the format of references are uniform. Moreover, please add some more references to support this study.

A: The references were checked and the format was corrected where necessary. Obviously, the list of references of the review cannot cover all the available literature on echinoderms. Following the suggestion of the reviewer I can add more references proposed by the reviewer, if there are any.

Reviewer 2 Report

It is suggested to add some content related to the composition and distribution of these fatty acids in structural lipids, such as the information on the correlation study of the distribution of these fatty acids in neutral lipids, phospholipids, and glycolipids.

Author Response

I would like to express my sincere gratitude to the Reviewer for the attention to my work and for the comments.

Q: It is suggested to add some content related to the composition and distribution of these fatty acids in structural lipids, such as the information on the correlation study of the distribution of these fatty acids in neutral lipids, phospholipids, and glycolipids.

A: Although there is no special section on the composition and distribution of the fatty acids in structural lipids, I touch this topic multiple times in the review, which covers the subject fully.

Reviewer 3 Report

The authors introduced a review about fatty acids in Echinodermata. Below are the comments.

1-      English editing is required.

2-      Figure 1: do you have the copyright or authorized permission for the presented photos ?

3-      Figure 2: please draw the fatty acids using specific programs like chemdraw and not use pictures.

4-      Please discuss and speculate the possible biosynthetic pathways for the amino acids in Echinodermata.

5-      Discuss the published GC-MS conditions and data for identification of these fatty acids.

Author Response

I would like to express my sincere gratitude to the Reviewer for the attention to my work and for the comments.

Q: 1. English editing is required.

A: English has been improved.

Q: 2. Figure 1: do you have the copyright or authorized permission for the presented photos ?

A: I have authorized permission for the presented photos. I thank Mr. K. Dudka for providing his photos of echinoderms in the section Acknowledgements.

Q: 3. Figure 2: please draw the fatty acids using specific programs like chemdraw and not use pictures.

A: I improved the Figure 2.

Q: 4. Please discuss and speculate the possible biosynthetic pathways for the amino acids in Echinodermata.

A: The review is devoted to fatty acids of echinoderms, I believe that the discussion of possible ways of biosynthesis of amino acids in echinoderms is unnecessary and untimely.

Q: 5. Discuss the published GC-MS conditions and data for identification of these fatty acids.

A: I share the opinion of the Reviewer that the accuracy and reliability of the data on the composition of fatty acids largely depends on the conditions of the GC analysis. That is why in the review I have emphasized that the variability of the results of different authors is often and largely determined by differences in the conditions of GC analysis and by the methods of fatty acid identification. However, due to the abundance of data discussed, it is technically not possible to give the analysis conditions.
